# What Is The Political Content in LLMs' Pre- and Post-Training Data?

## Abstract

Large language models (LLMs) are known to generate politically biased text, yet how such biases arise remains unclear. A crucial step toward answering this question is the analysis of training data, whose political content remains largely underexplored in current LLM research. To address this gap, we present in this paper an analysis of the pre- and post-training corpora of open-source models released together with their complete dataset. From these corpora, we draw large random samples, automatically annotate documents for political orientation, and analyze their source domains and content. We then assess how political content in the training data correlates with models' stance on specific policy issues. Our analysis shows that left-leaning documents predominate across datasets, with pre-training corpora containing significantly more politically engaged content than post-training data. We also find that left- and right-leaning documents frame similar topics through distinct arguments and that the predominant stance in the training data strongly correlates with models' political biases when evaluated on policy issues. Finally, pre-training corpora subjected to different filtering and curation procedures exhibit broadly similar characteristics with respect to their political content. These findings underscore the need to integrate political content analysis into future data curation pipelines as well as in-depth documentation of filtering strategies for transparency.

## 1 Introduction

Large language models (LLMs) and chatbots have rapidly gained widespread adoption, with leading platforms reaching over 700 million weekly active users,[1] and more and more people use AI systems as sources of factual information, commentary, and practical advice (Ouyang et al., 2023; Bassignana et al., 2025). When looking for political information, Luettgau et al. (2025) observe that not only around 13% of eligible voters may have used conversational AI to inform their electoral choices, but also that such use enhanced political knowledge to a degree comparable with web search. Beyond knowledge acquisition, Potter et al. (2024) and Salvi et al. (2025) show that LLMs can influence users' decision-making and political beliefs by acting as information intermediaries. This growing evidence raises the question of whether AI systems are politically impartial and what viewpoints are ultimately shown to users. Biases can become harmful when model outputs reinforce only a narrow set of viewpoints tied to specific groups in society, thereby granting these groups disproportionate influence over decision-making and undermining a core democratic principle – the preservation of a public sphere where diverse opinions can coexist (Balkin, 2017). However, research on political biases in LLMs remains surprisingly limited, especially when compared to the extensive body of work on biases related to gender and race (Garg et al., 2018; Savoldi et al., 2021).

Existing studies of political biases in LLMs have consistently suggested that models mostly reflect left and libertarian worldviews in their generated texts (Motoki et al., 2024; Hartmann et al., 2023; Röttger et al., 2024; Ceron et al., 2024). Other studies have observed that political alignment of models can be marginally influenced in the post-training phase with additional and targeted data regarding political content (Stammbach et al., 2024; Weeber et al., 2025). However, the origins of bias remain uncertain: do they stem mainly from patterns already present in pre-training data, or

---

[1] https://techcrunch.com/2025/08/04/openai-says-chatgpt-is-on-track-to-reach-700m-weekly-users

are they embedded in subsequent post-training stages? Answering these questions informs training practices, therefore allowing for the design of fairer models. In this paper, we present a deeper understanding of these questions taking on a data perspective. We investigate the prevalence of political content in the various training stages, the framing of this content, and its correlation with model behavior.

Despite broad consensus that a deep understanding of training data is essential for interpreting machine learning models, the contents of datasets used to train LLMs remain significantly underexplored. Although prior work has emphasized the importance of data transparency and scrutiny (Mitchell et al., 2019; Hutchinson et al., 2021) and called for greater attention to training data (Kaddour et al., 2023), research has focused disproportionately on model architecture and performance. A central obstacle is that most LLM providers disclose little to no detail about their training data, be it in open-weight models (e.g., Touvron et al., 2023; Yang et al., 2025) or closed models such as ChatGPT (Achiam et al., 2023), Claude (Anthropic, 2025), or Gemini (Comanici et al., 2025). Even when details about data mixtures are available, the sheer scale of these corpora and the practical challenges of storing and processing them hinder systematic investigation.

In this work, we introduce an actionable sampling-based framework to address such challenges. We investigate three open source models: OLMO2 (OLMo et al., 2024), Pythia (Biderman et al., 2023), and Falcon (Malartic et al., 2024) given their distinct pre-training datasets. However, we place special focus on OLMO2 given that it is the only model with different pre- and post-trained models available with the respective data. We extract representative samples from the pre- and post-training corpora to mitigate the computational challenges posed by their scale. These samples are classified as left-leaning, right-leaning, or neutral with a novel classifier explicitly validated for robust performance on out-of-domain data. This enables us to estimate both the overall level of political engagement in the training corpora and the political orientation in this subset. Then, we provide an in-depth, semi-automated analysis of content in the politically engaged texts from the training corpora, providing richer context for interpreting models' behavior. Finally, we apply the political worldview-probing methodology proposed by Ceron et al. (2024) to assess whether LLMs demonstrates tendencies congruent with the stance of the content from its training data.

Overall, we find that left-leaning documents consistently outnumber right-leaning ones by a factor of 2.3 to 12 across training datasets, with pre-training corpora containing about from 2.5 to 4 times more politically engaged content than post-training data. The framing of political topics also varies considerably: right-leaning labeled documents prioritize stability, sovereignty, and cautious reform via technology or deregulation, while left-leaning documents emphasize urgent, science-led mobilization for systemic transformation and equity. In addition to that, the source domains of pre-training documents also differ significantly, with right-leaning content containing twice as many blog posts and left-leaning content 3 times as many news outlets. We also observe a strong correlation of $r=0.87$ between the predominant stances in the training data and the models' behavior when probed for political bias on eight policy issues (e.g., environmental protection, migration, etc). Finally, our analysis across pre-training datasets from different model families indicates that variations in data curation and filtering strategies have minimal impact on the main properties of political content within the training corpora.

**Contributions.** To the best of our knowledge, this is the first study to analyze political content pre-training, mid-pre-training, and post-training datasets across models. Our contributions are threefold. First, we release a dataset of training documents automatically annotated as left-leaning, right-leaning, or neutral. Second, we provide a systematic analysis of source domains, topics, and framing strategies in the politically engaged portions across training datasets. Third, we examine how dominant views in the training data correlate with model behavior in terms of political biases across model families, focusing on fine-grained, politically salient policy issues.

## 2 RELATED WORK

Elazar et al. (2024) propose a search and count method to process and analyze the bulk of the pre-training data in an accessible way. Piktus et al. (2023a) make available a search tool to explore ROOTS, the dataset used to train the open source model BLOOM with fuzzy or exact matches, while Piktus et al. (2023b) propose a search engine to understand datasets before even using them

for training. Other works have instead proposed methods and analyzed the effect of data on model behavior, for example, in the context of fact checking (Akyurek et al., 2022), looked at the correlation between model performance and data quality (Longpre et al., 2024), and estimated how similar generated text is to the training data (McCoy et al., 2023).

Xu et al. (2025) is, to the best of our knowledge, the only work that investigates political content in pre-training data. They use Elazar et al.'s (2024) search tool to extract documents that contain terms related to cases from 32 U.S. Supreme Court cases on topics including abortion and voting rights. Our work differs from the approach of Xu et al. (2025) as we evaluate political content in broader policy issue categories more applicable for cross-country comparisons. Moreover, the pre-training data used in our analysis is sampled from DOLMA and DOLMINO, which contain several pre-training datasets. Finally, our work also sheds light on post-training, which is highly relevant to how models eventually behave in user interactions given the steerability of models in that phase (Stammbach et al., 2024; Weeber et al., 2025).

Previous work investigating political bias in LLMs can be found in the social sciences and computer science literature. Many previous studies conducted a descriptive analysis of the political leaning present in LLMs (e.g., Motoki et al., 2024; Hartmann et al., 2023; Rutinowski et al., 2024; Röttger et al., 2024; Ceron et al., 2024; Röttger et al., 2025). Overall, results suggest that large commercial LLMs have a consistent left-leaning bias, whereas political leanings for smaller open-source models seem to be more mixed and less consistent in their stance (Ceron et al., 2024).

Other work looked into the steerability of aligning large language models with specific political biases (Jiang et al., 2022; Feng et al., 2023; Stammbach et al., 2024) and found that it is possible to slightly change the political leaning, steering it to be less left-leaning. This also holds for non-US contexts, such as Swiss politics (Stammbach et al., 2024), and it has been observed that steering leaning in one language also impacts political beliefs in other languages (Weeber et al., 2025).

Another set of studies has focused on the impact of political leanings in LLMs on human political beliefs (Potter et al., 2024; Hackenburg & Margetts, 2024; Bai et al., 2025). They show that LLMs are (hidden) persuaders, as they can influence political beliefs and potentially impact voting behavior in humans. More broadly, these studies contribute to the literature on how LLMs can influence human decision-making (Stieger et al., 2021; Sharma et al., 2024). Taken together, such results, combined with the scale of LLM adoption and everyday usage, underscore the urgency of understanding where political bias in LLMs might stem from. To the best of our knowledge, this is the first study to address this by analyzing the political content of LLM training data across different stages, releasing annotated datasets, and correlating data characteristics to model behavior.

## 3 DATA

### 3.1 TRAINING DATA FROM LLMS

We analyze the training data from 3 models. OLMO2, FALCON and PYTHIA. We choose these three models because their training datasets do not overlap in any way. Table 4 in Appendix A.1 shows in details the overlap of datasets across models. Table 1, instead, shows the datasets analyzed in this study. DOLMA and DOLMINO are the datasets used in the pre-training and mid-pre-training stages of the OLMO2-BASE, respectively. As reported in OLMo et al. (2024), the stages differ in floating-point operations (FLOPs) and the data: pre-training relies on a broader, less filtered collection of web-scraped documents (DOLMA) and runs longer with 90% of training FLOPs, whereas mid-pre-training continues next-token prediction on a higher-quality, filtered dataset (DOLMINO) for 10% of training FLOPs. REFINEDWEB is used to pretrain FALCON whereas the deduplicated version of THEPILE is used to train PYTHIA.

SFT-MIX is the dataset used in the first post-training stage of the base model for supervised fine-tuning. This stage also relies on next-token prediction, but the inputs are reformatted using a chat template. During this phase, the model learns to answers a variety of traditional NLP tasks contained

Table 1: Training datasets from OLMO2 Pythia, and Falcon, with the number of sampled documents and their proportion classified as left- or right-leaning. The remaining documents in the classification results were classified as *neutral*.

| Documents for analysis | | | | Classification results | |
|---|---|---|---|---|---|
| **Dataset name** | **Total size** | **Training and model** | **# Docs** | **# (%) Left** | **# (%) Right** |
| **DOLMA**
OLMO-MIX-1124 | 22.4TB | Pre-training
OLMO-2-1124-13B | 299,915 | 12,790 (4.2%) | 4,644 (1.5%) |
| **DOLMINO**
DOLMINO-MIX-1124 | 5.14TB | Mid-pre-training
OLMO-2-1124-13B | 200,000 | 10,537 (5.2%) | 3,374 (1.6%) |
| **REFINEDWEB**
FALCON-REFINEDWEB | 1.68TB | Pre-training
FALCON-11B | 200,000 | 6,089 (3,0%) | 2,676 (1.3%) |
| **THE PILE**
THE_PILE_DEDUPLICATED | 451GB | Pre-training
PYTHIA-12B-DEDUPED | 200,000 | 7,807 (3,9%) | 3,060 (1.5%) |
| **SFT-MIX**
TULU-3-SFT-OLMO-2-MIXTURE | 1.41GB | Supervised finetuning (SFT)
OLMO-2-1124-13B-SFT | 193,447 | 2,863 (1.5%) | 242 (0.12%) |
| **DPO-MIX**
OLMO-2-1124-13B-PREFERENCE-MIX | 1.46GB | Direct Preference Optim. (DPO)
OLMO-2-1124-13B-DPO | 313,609 | 5,792 (1.8%) | 920 (0.29%) |

in the dataset in dialogue turns. Finally, DPO-MIX is the dataset used in the second post-training phase, where the model learns human preferences with DPO.[2]

**Sampling pre-training data.** Since investigating the content of the whole DOLMA and DOLMINO datasets is not feasible due to computational resource constraints, we resort to sampling to retrieve a representative set of documents for our analysis of the pre-training data. For that, we first fix a sample size of $N$ documents and then use the classic version of the reservoir sampling algorithm (Vitter, 1985). This extracts uniform random samples from the base dataset in an online fashion, Excluding the subset focused on programming languages and code from our analysis. $N$ is set to first 100k and then 200k documents for DOLMA and 200k for DOLMINO and SFT-MIX. In the last dataset, the assistant replies in conversation chains, so the documents are constructed by concatenating the turns together into a single document. We include all the documents from DPO-MIX because the number of documents is on the order of 300k, considerably lower than the other datasets. To ensure quality in the subsequent analysis, we restrict our study to the English subset. According to the metadata, DOLMINO only contains English documents, most probably because of the hand-picked source domains for high-quality while we remove 85 non-English documents. We automatically identify the non-English documents from the post-training datasets using a fastText-based language classifier. As a result, for DPO-MIX we include 313,609 out of 378,339 documents, and for SFT-MIX we include 193,447 out of 939,344 documents. THEPILE and REFINEDWEB are English datasets, so we directly sample 200K documents from them.

## 3.2 VALIDATION DATASETS

We rely on several datasets to validate the classifiers which aid in our analysis of the training documents. Given that many documents from the pre-training datasets come from news outlets and archives, the validation sets should resemble this type of data. The first is NEWSLEAN, this is the only dataset with news articles that was manually annotated per article, and not heuristically annotated by automatically deducing labels from the news outlet. It is a proprietary news article dataset with documents from 110 news outlets from the US which were manually annotated with labels on a scale from very left to very right. We map these labels to *left*, *right*, and *neutral*. NEWSLEAN contains a total of 4,434 news articles (details about the annotations in A.3). 91 outlets present in the news article dataset overlap with the outlets in the source domains of our sample of the pre-training data, indicating high similarity with the training data on which we implement the classifier.

The second dataset is US CONGRESS, consisting of two sets of speeches from the US Congress annotated as left, moderate right, and extreme right by Seow (2025). This dataset was selected because of the clear ideological polarization present in congressional speeches, which provides a separate evaluation setting compared to journalistic text. Either set contains 600 speeches from 8

---

[2]We do not analyze the data used in the subsequent post-training phase of the Instruct version, as it mainly consists of mathematical content or repeated material from SFT-MIX, cf. `https://huggingface.co/allenai/OLMo-2-1124-13B-Instruct`

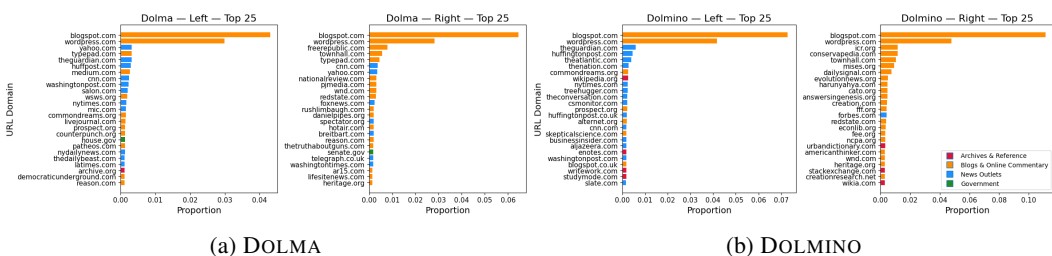

(a) DOLMA

(b) DOLMINO

Figure 1: Relative number of the distribution of source domains in the pre-training datasets.

and 10 different speakers, respectively, evenly balanced between the three classes. The third dataset is PROBVAA (Ceron et al., 2024), containing statements extracted from voting advice applications from 7 EU countries which annotations for stances towards policy issues (details in Appendix A.3.1).

Additionally, we build a last dataset with stance labels STANCEPOL. This consists of 200 documents extracted from the pre-training data which are annotated for stance towards the same policy issues present in PROBVAA. The documents from STANCEPOL are then additionally mapped to left and right labels based on the stance labels (see all details in Appendix A.3.1).

While machine learning papers often measure validity as a single performance number (e.g., F1 or accuracy), political scientists take a more holistic approach. Goet (2019) introduces face validity (superficially, does the measure intuitively seem convincing), convergent validity (does the measure correlate with performance numbers and theory), and construct validity (does the measure have predictive power). We replicate this approach. For face validity, we show 5 random text samples classified as left-leaning, neutral, and right-leaning in Appendix A.4.2. For convergent validity, we refer to the classification results across multiple datasets introduced in this section and topic modeling results in Section 5.2. For construct validity, we show correlations of pre-training bias and model stance in Section 6.1.

## 4 LEFT–RIGHT CLASSIFICATION

Previous studies report that LLMs reflect more left-leaning views (Motoki et al., 2024; Ceron et al., 2024). We make the assumption that such behaviors stem from documents in the training data that contain political content and express a leaning towards the left. Thus, a fundamental step of our pipeline is to first identify documents that contain such political content and their corresponding political leaning. For that, we evaluate several methods for classifying documents for left-, neutral-, and right-leaning documents.

In our setup, neutral documents are documents that either lack political content altogether or do not express an evident right- or left-leaning stance. We acknowledge that this is a simplification: (i) a centrist political position is an ideology on its own right and that (ii) politically "neutral" texts are not really neutral. In principle, we agree on both points; however, given current data availability, training a reliable classifier that captures these nuances is currently not feasible.[3] We therefore assume that, despite their lower prevalence, more engaged documents exert a stronger overall influence on model biases.

**Classification pipeline.** We evaluate large LLMs from the LLaMA, Gemma, and Qwen families and several prompt instructions in the news article dataset in a classification scenario between *left*, *right*, and *neutral*. Our best performing model and prompt instruction was the quantized variant LLAMA3.3-70B-4BIT, reaching a macro-F1 score of 71%. We also evaluated fine-tuned models such as RoBERTa-based models used by Nikolaev et al. (2023) and fine-tuned ModernBERT (Warner et al., 2025), but they did not perform well in out-of-domain documents, standing roughly 15 points behind in performance (detailed results of the validation are in Appendix A.4).

---

[3]Preliminary topic-modeling analysis of "neutral" documents shows that they are largely non-political in content. Cf. results in Appendix A.5

Then, we further validate the robustness and generalizability of the best performing classifier in two additional datasets: the left and right labels of POLSTANCE and US CONGRESS. In the former, it reaches a performance of 82% macro-F1, showing solid in-domain performance. In the latter, the classifier achieves macro-F1 scores of 73% and 72% in the two sets, confirming its robustness in accurately detecting the leaning of political content (see detailed results in Appendix A.4).

**Political content in training data.** The best-performing model and prompt combination is used to classify all sampled documents for our analysis, as shown in Table 1. Among the pretraining datasets, DOLMINO has the highest share of politically classified documents, with 1.6% right-leaning and 5.2% left-leaning, while REFINEDWEB has the lowest with 3% left- and 1,3% right-leaning documents. Second and third are DOLMA and THEPILE with similar percentages. In the post-training datasets, SFT-MIX has the lowest, with 0.12% right- and 1.5% left-leaning documents while DPO-MIX has slightly more with 0.29% right- and 1.8% left-leaning documents. Results suggest that even though the proportion of documents containing political content is low, the percentage of left-leaning documents is consistently higher across datasets, ranging from 2.8 times more left-leaning content in DOLMA up to 12.5 times more left-leaning content in the SFT-MIX. The low proportion of politically engaging texts in the latter is explained by the fact that the data mostly comes from supervised fine-tuning tasks. This finding also aligns with the results of political-bias studies showing that the generated content of models reflects a more left-leaning orientation (Ceron et al., 2024; Röttger et al., 2024; Motoki et al., 2024).

## 5 POLITICAL CONTENT ANALYSIS

For the remainder of this paper, we restrict our analysis to documents classified as exhibiting either a left- or right-leaning orientation across all datasets. Any reference to the pre-training or post-training corpus should be understood as referring to this politically engaged subset. Given the similarity in results, results from REFINEDWEB and THEPILE regarding this section are in Appendix .

### 5.1 SOURCE DOMAINS

Analyzing the source domains of the pre-training data, which consist of web-scraped documents unlike post-training datasets, provides insights into the origins of political content and serves as a proxy for data quality and factuality. We extract source domains from document metadata in DOLMA and DOLMINO by preprocessing URLs. Figure 1 shows the relative distribution of source domains in the pre-training (DOLMA) and mid-pre-training (DOLMINO) data. The most notable observation is that political content in the training corpus mostly comes from blog posts: BlogSpot and WordPress contribute to the largest number of documents across all datasets, and TypePad is very prominent in DOLMA. Among the top 25 source domains, blogs account for a larger share of right-leaning documents, with 17% in DOLMA and 21% in DOLMINO, compared to 12% and 7% in the left-leaning documents, respectively. In contrast, the left-leaning documents have a higher proportion of highly-ranked news outlets, with 11% in DOLMA and 14% in DOLMINO, compared to 7% and 1% in the right-leaning documents, respectively.

The major sources of left-leaning texts in both datasets are the leading news outlets The Guardian and HuffPost (formerly The Huffington Post). Prominent US newspapers including The New York Times and The Washington Post also contribute, though to a lesser extent. Less prominent outlets, including Salon, Business Insider, The Conversation, and the L.A. Times, also appear among the sources, together with activist and issue-focused media like Common Dreams and Tree Hugger, though these account for only a very small share.

The right-wing subset of DOLMA also has some content from traditional right-wing media, such as National Review (US) or The Telegraph and The Spectator (UK), but they are less prominent than younger online-only communities and media, such as FreeRepublic and Townhall. In DOLMINO, the right-leaning content consists almost entirely of blog posts. Among these, Townhall is also present but less prominent than the Institute for Creation Research (ICR) and Conservapedia, two fundamentalist Christian resources. Overall, the right-wing sources are more varied, recent, extreme, and narrowly focused than left-wing ones – which, in DOLMINO, also notably includes Wikipedia. Other sources beyond social media and news outlets also appear in the data, primarily information repositories such as Archive.org or government websites like House.gov, but their overall contribu-

**Dolma**

| Political leaning Label | left | right |
|---|---|---|
| Climate Change and Energy | 0.31 | 0.12 |
| Christianity and Faith | 0.09 | 0.32 |
| UK Politics | 0.15 | 0.09 |
| LGBTQ+ Rights | 0.13 | 0.05 |
| Gender and Relationships | 0.07 | 0.08 |
| Indian Politics | 0.06 | 0.07 |
| Reproductive Rights | 0.05 | 0.13 |
| Gun Control Debate | 0.02 | 0.13 |
| Israeli-Palestinian Conflict | 0.05 | 0.07 |
| Contemporary Art and Culture | 0.07 | 0.01 |

**Dolmino**

| Political leaning Label | left | right |
|---|---|---|
| Religion and Evolution | 0.20 | 0.55 |
| Animal Rights and Food | 0.19 | 0.04 |
| Climate Change | 0.10 | 0.09 |
| Nuclear Arms and Geopolitics | 0.08 | 0.05 |
| Education Policy | 0.08 | 0.05 |
| Monetary Policy and Banking | 0.06 | 0.08 |
| Abortion and Reproduc Rights | 0.06 | 0.06 |
| Latin American Politics | 0.08 | 0.03 |
| Electoral Reform | 0.07 | 0.04 |
| Iraq War | 0.08 | 0.02 |

**SFT-mix**

| Political leaning Label | left | right |
|---|---|---|
| Climate and Sustainability | 0.19 | 0.22 |
| LGBTQ+ Inclusive Education | 0.15 | 0.00 |
| Literary Themes and Narratives | 0.11 | 0.36 |
| Hate Speech Restrictions | 0.10 | 0.06 |
| Body Image Positivity | 0.10 | 0.00 |
| Homelessness and Housing | 0.08 | 0.03 |
| Gender Equality in Workplace | 0.08 | 0.00 |
| Educational Inequality | 0.07 | 0.17 |
| Mental Health Policy | 0.06 | 0.17 |
| Disability Inclusion and Respect | 0.06 | 0.00 |

**Dpo-mix**

| Political leaning Label | left | right |
|---|---|---|
| Diverse Storytelling | 0.16 | 0.06 |
| Community Speech and Politics | 0.12 | 0.18 |
| Promoting Respectful Dialogue | 0.15 | 0.02 |
| Gender Equality | 0.14 | 0.03 |
| LGBTQ+ Gender Identity | 0.12 | 0.02 |
| Marxism and Capitalism | 0.07 | 0.18 |
| Lenin and Soviet Law | 0.06 | 0.15 |
| Climate Change Impacts | 0.08 | 0.03 |
| Inclusive Education | 0.08 | 0.03 |
| East Asian Empire Restoration | 0.03 | 0.30 |

Figure 2: Proportion of documents belonging to the 10 most dense clusters per left and right documents. Same colors in the y-ticks point to the similarity in topics across datasets.

tion is marginal. In DOLMA, Archives & Reference account for only 1% of left-leaning documents, while in DOLMINO they represent 4% and 3% of left- and right-leaning documents, respectively. The slightly higher proportions in DOLMINO are consistent with its stricter data quality filtering.

## 5.2 TOPIC ANALYSIS

To complement the document source perspective, we next turn to the substantive content of the documents by examining their main topics. This makes it possible to identify whether left- and right-leaning texts engage with the same issues, and more importantly, to uncover systematic differences in how those issues are framed. To extract topics, we automatically cluster the documents using the BERTopic model (Grootendorst, 2022) based on document representations obtained with ALL-MPNET-BASE-V2, with truncation to the maximum number of tokens. Datasets are clustered independently to understand what topics emerge from each dataset and how they differ. Then, we use GPT-4.1-NANO for generating the labels of the topic clusters (implementation details are provided in Appendix A.7).

**Topic distribution.** Figure 2 shows the proportion of documents belonging to the 10 densest clusters for each dataset. The topic of *Climate Change* occurs across all datasets. Other topics, such as *Religion and Christianity* and *Reproductive Rights*, are often present in the pre-training data, whereas *LGBTQ+ Rights* occur in high frequency in all datasets except DOLMINO, where they rank only 12th as shown in Appendix A.7. We observe that the post-training datasets contain more "normative" topics than the pre-training datasets, which is to be expected since the post-training phase aligns the models for harmlessness and helpfulness. For example, both SFT-MIX and DPO-MIX include topics such as *Inclusive Education*, *Gender Equality*, and *Hate Speech Restrictions* in their top-10 clusters.

**Political messaging.** To analyze differences in the messages conveyed by left- and right-leaning documents within clusters, we summarize the main message of the documents using GPT-5. We keep at most 300 documents per topic cluster and truncate the documents to contain at most 300 tokens. We join the documents into a single string and prompt the model to "summarize the main message of the documents in a couple of sentences". For the analysis, we selected the 5 most dense clusters among the left- and the right-leaning documents. Table 17 in the Appendix shows one selected cluster per dataset (the remaining are found in Appendix A.7).

Based on these content summaries, we conduct a high-level analysis that goes beyond the surface descriptions to highlight how left- and right-leaning documents differ not only in policy preferences or historical interpretation but also in the framing of legitimate concerns and authoritative knowledge. On *Climate and Sustainability*, the right documents foreground economic stability, sovereignty, and skepticism of rapid regulatory change, often invoking technology or deregulation as pragmatic solutions. The left documents, by contrast, stress urgency, scientific authority, and mobilization across actors toward systemic transformation and equity. On *East Asian Empire Restoration*, the right cluster reimagines history through revisionist, restorationist, and authoritarian narratives, while the left cluster responds by demystifying propaganda, emphasizing historical atrocities, and centering justice for marginalized groups. Regarding *Christianity and Faith*, the right texts reinforce biblical authority, evangelism, and conservative moral order, while the left clusters critique institutional abuses and advocate reform, inclusion, or secular humanist ethics. Finally, on *Animal Rights and Food*, the right classified documents emphasize personal responsibility, stewardship, and market-

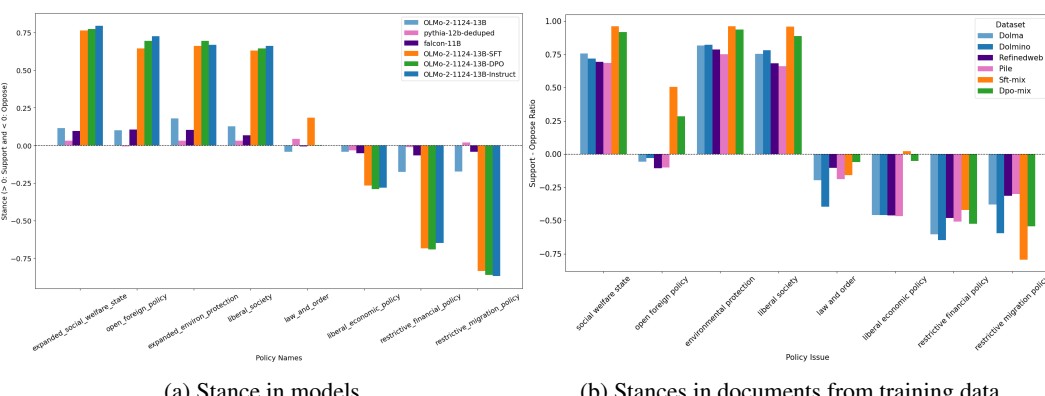

(a) Stance in models        (b) Stances in documents from training data

Figure 3: Stances of the models at different training stages and in the documents from the training data across policy issues. Colors of the bars match models and datasets used to train them at that stage. DOLMA and DOLMINO are in blue because they are both used in the base model.

based conservation, whereas the left ones draw attention to structural harms, suffering, and ecological impacts and promote the advancement of plant-based diets. Overall, the results show that while left- and right-leaning documents often address the same topics, they frame them through different arguments and emphases.

## 6 POLICY ISSUE ANALYSIS

In this section, we examine how political biases in model behavior correlate with the training data, with the aim of shedding light on which stage of the training process such biases are introduced and reinforced. Political bias in models refers to when they consistently support or oppose viewpoints towards a policy issue (e.g., support for an expanded social welfare state).

### 6.1 POLITICAL BIASES IN MODELS

To compute model stance on policy issues, we follow a similar approach and the dataset ProbVAA provided by Ceron et al. (2024); details are provided in Appendix A.8.

**Model's behavior.** Figure 3a shows the results of the political bias analysis. All base models reflect a weaker signal in bias direction. This is attributed to the lower consistency in the answers generated by the base model (cf. consistency effects in Appendix A.8). However, the bias direction is very similar across models given that they all support or oppose the same policy issues except for PYTHIA-12B which has a slight tendency to support *Restrictive Migration Policy* while the other models oppose it. In terms of policy issue, the only except is *Law and Order* where the stance signal is much weaker across models in comparison with the other policy issues, suggesting that models do not have a strong preference towards this issue.

The political viewpoints reflected in the answers of the models is very similar across all model checkpoints and they are mostly in line with the viewpoints emphasized by the left-leaning agenda. Their answers reflect a neutral positioning towards *Law and Order*. Their generated answers are more in favor of the left-leaning policies (*Expanded Social Welfare State*, *Open Foreign Policy*, *Expanded Environmental Protection*, and *Liberal Society*), while their answers tend to be against *Liberal Economic Policy* and *Restrictive Migration Policy*. The answers also largely disagree with *Restrictive Financial Policy*, which can be either left or right depending on the specific policy (e.g., the left may support tax increases to reduce deficits, while the right may aim to minimize debt and deficits). Notably, the same trends are observed in different parameter size models from OLMO2 (OLMO2-7B and OLMO2-32B) and in other open source models such as Apertus-8B (Hernández-Cano et al., 2025) and SmolLM3-3B (Bakouch et al., 2025) (see results in Appendix A.8).

Table 2: Pearson $r$ correlation between the model's stance at different training stages and the stances of the training documents. * indicates p-val<0.05.

| Model | Dataset | $r$ |
|---|---|---|
| OLMo-2-13B | Dolma | 0.88* |
| OLMo-2-13B | Dolmino | 0.91* |
| OLMo-2-13B-SFT | SFT-mix | 0.93* |
| OLMo-2-13B-DPO | DPO-mix | 0.93* |
| OLMo-2-13B-Instruct | DPO-mix | 0.92* |
| falcon-11B | RefinedWeb | 0.82* |
| pythia-12B-deduped | Pile | 0.63 |
| Average Pearson's $r$ | | 0.87 |

Table 3: Pearson correlations between pretraining datasets.

| Dataset | Dataset1 | $r$ |
|---|---|---|
| RefinedWeb | Dolma | 0.99* |
| RefinedWeb | Dolmino | 0.97* |
| Pile | Dolma | 0.99* |
| Pile | Dolmino | 0.98* |
| Dolma | Dolmino | 0.99* |

## 6.2 CORRELATION BETWEEN BIASES IN THE TRAINING DATA AND IN THE MODELS

In this section, we aim to understand whether there is a correlation between the stance of the models in the investigated policy issues, as observed in the previous section, and the stances in training data. To that end, we analyze the stance of documents towards the same policy issues as in § 6.1.

**Classification pipeline.** We evaluate zero-shot prompts for a multi-label stance classification task where the objective of the classifier is to identify support or opposition towards policy issue(s) in a given document. For this, we use META-LLAMA-3.1-70B-BNB-4BIT, with the best results obtained from a Chain-of-Thought prompt including explicit guidelines. The results are evaluated against the ground truth labels of our dataset STANCEPOL. The average macro-F1 across policy domains is 65% with scores varying between 83% for *Restrictive Migration Policy* to 53% for *Law and Order*. Although average performance is modest, it is sufficient to capture reliable coarse-grained patterns in the training data across policy domains grounded in political science. More details about the results and experiment are in Appendix A.9.

**Stances analysis in training documents.** We extract the stance of documents towards the 8 policy issues with our stance classifier and use these labels to compute the general stance of the training data. We compute the final stance per policy issue as $Spol = \frac{S-O}{S+O}$ where $S$ is the total number of documents supporting and $O$ is the total number of documents that opposing the policy issue. Figure 3b shows the results for this computation. The stance is very similar across all pre- and post-training datasets with little variance across all policy issues except for *Open Foreign Policy* and *Liberal Economic Policy*. In *Law and Order*, DOLMINO has a stronger stance in comparison with the other datasets, while SFT-MIX shows a stronger stance than the other datasets in *Restrictive Migration Policy*. The pre-training datasets have a strong stance against *Liberal Economic Policy* in comparison with the post-training datasets. In *Open Foreign Policy* instead, the pretrained datasets have the opposite stance from the post-training datasets which are more in favor of the policy. This is the only policy issue that does not correlate well with the results of the models, given that the base model expresses support in this policy (as seen in Figure 3a).

Overall, the stances of the training documents are strongly correlated with those of the models, with an average Pearson $r$ of 0.87 across model–dataset pairs as seen in Table 2). Moreover, Pearson correlations do not increase considerably with following training stages. Among the OLMO2 models, $r$ varies from 0.88 between DOLMA and OLMO2-13B-BASE to 0.93 between DPO-MIX and OLMO2-13B-DPO, allowing us to hypothesize that political biases are encoded already in the pretraining stage. The base models FALCON-11B and PYTHIA-12B have a slightly lower correlation, but this is expected given that these base models are very inconsistent and have a weaker signal in the model's behavior experiments. These results provide compelling evidence that model behavior is strongly influenced by the political viewpoints embedded in the training data. Finally, Table 3 shows the very high correlation between the political stances found in the pretraining datasets, with DOLMA being the most correlated to REFINEDWEB and THEPILE ($r$=0.99), but also very similar to DOLMINO ($r$=0.97) and ($r$=0.98) respectively.

## 7 DISCUSSION

Our results point to several factors that contribute to the observed biases. Regarding data, in the OLMO2 models which have more training stages available, we notice that DOLMA and DOLMINO exhibit a markedly higher proportion of politically loaded content, compared to data used at later stages, particularly left-leaning documents (§ 4). Furthermore, political biases reflecting support for the left-leaning agenda are also prominent in the base models (7B, 13B, and 32B). Together, these observations suggest that political biases are already formed during the pre-training phase. This is in contrast to what has been previously hypothesized (Hagendorff, 2025). The fact that fine-tuned models sometimes demonstrate markedly biased behavior could be because biases are either reinforced in the post-training stages or that models become more consistent in their generated answers. We cannot disentangle stance reinforcement from improved consistency with the current evaluation setup. This interpretation could also help explain why recent work (Weeber et al., 2025; Fulay et al., 2024) reports only limited success in aligning models to different political leaning through post-training interventions or fine-tuning reward model for truthfulness achieve only limited success: when strong imbalances are already encoded in pre-training data, as our study shows, alignment can only partially steer models' political biases, rather than fully reverse them. It also validates the findings from Xu et al. (2025) which show a strong correlation between political biases related to the US court issues and the training data and from Röttger et al. (2025) where writing assistant have consistent biases towards some issues even though they are prompted to write from a different perspective.

Finally, Table 4 shows that the largest amount of the pretraining data comes from web-scraped documents from datasets such as C4 (Raffel et al., 2020), DCLM (Li et al., 2024), and FineWeb (Penedo et al., 2024). Even though these datasets originate from the CommonCrawl archive, the question of whether the filtering and curation influences the stance of the political content in training data remained open. However, our analysis across models with non-overlapping datasets (e.g. OLMO2, Falcon and Pythia, cf. A.1) reveals high similarity in the distribution of source domains, topics, and political stance in documents (cf. A.10 and Figure 3b) across datasets, suggesting that the data filtering and curation do not influence the main properties of the political content present in pretraining data. This also explains the high similarity in political biases across models from different families observed in our results and in previous studies as well (Ceron et al., 2024; Weeber et al., 2025).

## 8 CONCLUSION

Our study shows that left-leaning documents consistently outnumber right-leaning ones in both pre- and post-training data across datasets, with political content being far more prevalent in the pre-training stage. Our analysis further reveals systematic differences in source domains and topical framing: right-leaning material includes a larger share of blogs and emphasizes stability or sovereignty, while left-leaning material contains more established news outlets and highlights urgency, science, and systemic change. We also find a strong correlation between the stances expressed in model outputs and those predominant in the training data. By addressing political bias through the lens of data, we offer a robust reproducible methodology for analyzing how biases in the data correlate with biases reflected in LLMs. This perspective underscores the need for greater transparency in dataset curation and motivates development of strategies to mitigate politically biased content.

Future research could explore strategies to mitigate political bias in LLMs for more impartial models. For example, O'Brien et al. (2025) suggest that an effective method to reduce model knowledge about biothreats is to remove such data from the pre-training corpus altogether. This might provide an actionable solution for removing political bias in LLMs going forward: pre-filter pre-training data and remove heavily politicized text, which plausibly reduces political leanings of LLMs, and soften the impact on political views reflected in models.

## ETHICS STATEMENT

Our study focuses exclusively on the English documents from the OLMO2 family, as this is the only large-scale model released together with its full pre-training and post-training corpora. While this enables our analysis, it limits generalizability. However, given that most LLM providers rely

on similarly web-scraped corpora, our findings plausibly extend to other models unless substantially different filtering practices are applied. Ultimately, our analysis reveals correlations rather than causal mechanisms; we find strong evidence that biases originate primarily in pre-training and might be only reinforced in post-training. Further empirical research is necessary to validate these hypotheses.

The results of our study raise both ethical concerns and societal implications. Since LLMs are increasingly deployed in contexts ranging from education and information seeking to decision-support systems, the systematic bias toward particular political leanings can inadvertently influence public opinion or under-represent opinions from certain groups. Such risks are exacerbated by the content present in the training corpora and in the training strategies used in models. The ability to manipulate models' political leanings can, in the hands of ill-intentioned actors, serve malicious purposes to deliberately bias LLMs, in order to automate the generation of persuasive narratives, misinformation, or propaganda at scale. Moreover, outputs of LLMs that disproportionately align with one side of the political spectrum could deepen polarization, undermine trust in democratic institutions, or reinforce systemic inequities, particularly harming already vulnerable populations. However, we believe this type of research has a greater long-term benefit for society. On the one hand, it is essential for developing fairer and more impartial models, and on the other hand, it is crucial for understanding how malicious models work too.

Moreover, advancing research in this area can provide valuable insights for policymakers in the regulation of AI systems, as a deeper understanding of their mechanisms facilitates the development of more effective regulatory frameworks. For example, given the importance of pre-training data in models' knowledge, policymakers may demand a greater level of detail in the documentation of the data sources used for pre-training LLMs, and in particular, explanation of specific choices taken when curating training data.

Finally, our findings underscore the importance of dataset transparency and therefore accountability. For example, making design choices more explicit holds AI providers accountable for what they do. We aimed to contribute to the responsible research trajectory by empirically correlating political biases with pre-training data, thereby informing future efforts toward more pluralistic, transparent, and trustworthy AI systems.

## REPRODUCIBILITY STATEMENT

All the code and data are anonymously available at this link `https://osf.io/jyru2/?view_only=0cf00f3aeb2c4d56aff4efd2f5c3d203`. Appendix A.2 shows details of the random sampling approach. The details of data and annotations executed in the project are in Appendix A.3. Appendix A.4 contains details of the models, prompts, results, and validation of the political leaning classification. Appendix A.5 presents details of the documents classified as neutral. The token distribution of the documents can be found in Appendix A.6. Further results on the topic modeling analysis are in Appendix A.7. The methods and further results regarding model behavior in terms of political biases are in Appendix A.8. Finally, a description of the methods for classifying the stance of documents and results of other OLMO2 models can be found in Appendix A.9.

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

## A  APPENDIX

### A.1  DATASET OVERLAP ACROSS MODELS

Table 4 shows the pretraining datasets used in open weight models. All of them are also open source models except for Falcon.

### A.2  RANDOM SAMPLING

To check how representative our random sample is, we run the sampling algorithm on the DOLMA dataset twice. $N$ is set to 100k and then 200k. We then run our political leaning classifier on both

| Model | OLMO2 | OLMO3 | Marin-8B | SmolLM | Apertus | Falcon1/2 | Pythia | EuroLLM |
|---|---|---|---|---|---|---|---|---|
| Dolma1 | ✓ | ✗ | ✓ | ✗ | ✗ | ✗ | ✗ | ✗ |
| Dolmino1 | ✓ | ✗ | ✓ | ✓ | ✗ | ✗ | ✗ | ✗ |
| Dolma2 | ✗ | ✓ | ✗ | ✗ | ✗ | ✗ | ✗ | ✗ |
| Dolmino2 | ✗ | ✓ | ✗ | ✗ | ✗ | ✗ | ✗ | ✗ |
| DCLM-Baseline | ✓ | ✗ | ✓ | ✓ | ✗ | ✗ | ✗ | ✗ |
| DCLM-Edu | ✗ | ✗ | ✗ | ✗ | ✓ | ✗ | ✗ | ✗ |
| Overlap DCLM | ✓ | ✗ | ✓ | ✓ | ✓ | ✗ | ✗ | ✗ |
| FineWeb | ✓ | ✗ | ✗ | ✗ | ✗ | ✗ | ✗ | ✗ |
| FineWeb-2 (multilingual) | ✗ | ✗ | ✗ | ✗ | ✓ | ✗ | ✗ | ✗ |
| FineWeb-2-HQ | ✗ | ✗ | ✗ | ✗ | ✓ | ✗ | ✗ | ✗ |
| FineWeb-Edu | ✗ | ✗ | ✗ | ✓ | ✓ | ✗ | ✗ | ✓ |
| FineWeb-HQ | ✗ | ✗ | ✗ | ✓ | ✓ | ✗ | ✗ | ✗ |
| Overlab FINEWEB | ✓ | ✗ | ✗ | ✓ | ✓ | ✗ | ✗ | ✗ |
| Pile | ✗ | ✗ | ✗ | ✗ | ✗ | ✗ | ✓ | ✗ |
| CommonCrawl | ✗ | ✓ | ✗ | ✗ | ✗ | ✗ | ✗ | ✗ |
| Wikipedia | ✓ | ✓ | ✓ | ✓ | ✗ | ✗ | ✗ | ✓ |
| ArXiv | ✓ | ✓ | ✓ | ✓ | ✗ | ✓ | ✗ | ✗ |
| Books | ✓ | ✗ | ✗ | ✗ | ✗ | ✓ | ✗ | ✓ |
| peS2o | ✓ | ✗ | ✓ | ✓ | ✗ | ✗ | ✗ | ✗ |
| FLAN | ✓ | ✗ | ✓ | ✓ | ✗ | ✗ | ✗ | ✗ |
| Nemotron-CC | ✗ | ✗ | ✓ | ✗ | ✗ | ✗ | ✗ | ✗ |
| Proof-Pile-2 | ✗ | ✗ | ✓ | ✗ | ✗ | ✗ | ✗ | ✗ |
| SmolLM corpus | ✗ | ✗ | ✗ | ✓ | ✗ | ✗ | ✗ | ✗ |
| RefinedWeb | ✗ | ✗ | ✗ | ✗ | ✗ | ✓ | ✗ | ✗ |
| Reddit | ✗ | ✗ | ✗ | ✗ | ✗ | ✓ | ✗ | ✗ |
| HackerNews | ✗ | ✗ | ✗ | ✗ | ✗ | ✓ | ✗ | ✗ |
| Multilingual data | ✗ | ✗ | ✗ | ✗ | ✗ | ✓ | ✗ | ✗ |
| Scientific PDFs | ✗ | ✓ | ✗ | ✗ | ✗ | ✗ | ✗ | ✗ |
| HPLT (MLingual) | ✗ | ✗ | ✗ | ✗ | ✗ | ✗ | ✗ | ✓ |
| MADLAD-400 (MLingual) | ✗ | ✗ | ✗ | ✗ | ✗ | ✗ | ✗ | ✓ |
| CulturaX (MLingual) | ✗ | ✗ | ✗ | ✗ | ✗ | ✗ | ✗ | ✓ |
| mC (MLingual) | ✗ | ✗ | ✗ | ✗ | ✗ | ✗ | ✗ | ✓ |
| Apollo | ✗ | ✗ | ✗ | ✗ | ✗ | ✗ | ✗ | ✓ |
| Cosmopedia | ✗ | ✗ | ✗ | ✓ | ✗ | ✗ | ✗ | ✓ |
| RedPajama-Data-v2 | ✗ | ✗ | ✗ | ✗ | ✗ | ✗ | ✗ | ✓ |

Table 4: Comparison of pretraining datasets used across models. Mlingual means that the dataset has been built to be multilingual.

sets and check the proportion of left- and right-leaning documents. In 100k sample, the proportion of left- is 4,1% and right-leaning documents is 1,5% while in the 200k sample, the proportion is 4,3% of left and 1,6% right-leaning documents. This stability confirms the representativeness of our random sampling approach in terms of political content.

## A.3 DATA

**News article dataset.** This proprietary dataset comprises 4,434 articles from 110 US American news outlets published in 2019. Each news article was first manually annotated by 2 expert annotators recruited on Upwork. The requirement was to hold a postgraduate degree in relevant subjects (e.g., political science). After that, a third annotator, who is an undergraduate student, adjudicated the disagreements between the two expert annotators. The annotators labeled the articles with labels ranging from 1 to 7 with 1 being very left and 7 being very right. We mapped the labels between 3,5 and 4,5 as neutral, between 1 and 3 as left and between 5 and 7 as right. They were given the instruction to consider partisan use of language, political alignment, and the issues covered in the articles (e.g. more left-leaning or right-leaning covered issues).

We extracted from this news article validation set the source domains and we found that 91 source domains overlap with the source domains from DOLMA and DOLMINO. This illustrates the similarity between our validation set and the training data used in our analysis.

Table 5: Cohen's kappa between 2 annotators in the task of annotating the stance of documents towards policy domains.

| Policy issue | Cohen's $\kappa$ |
|---|---|
| restrictive-migration-policy | 0.81 |
| expanded-environ-protection | 0.72 |
| open-foreign-policy | 0.69 |
| expanded-social-welfare-state | 0.65 |
| law-and-order | 0.60 |
| liberal-society | 0.47 |
| restrictive-financial-policy | 0.43 |
| liberal-economic-policy | 0.39 |
| Average Cohen's $\kappa$ | 0.59 |

Overlapping web domains: theatlantic.com, thedailybeast.com, inquirer.com, clickondetroit.com, dallasnews.com, patch.com, nola.com, syracuse.com, nj.com, cincinnati.com, freep.com, cleveland.com, twitchy.com, nationalreview.com, breitbart.com, indystar.com, newsweek.com, foxbusiness.com, seattle-times.com, buffalonews.com, staradvertiser.com, latimes.com, bostonglobe.com, businessinsider.com, news-day.com, washingtontimes.com, detroitnews.com, hotair.com, foxnews.com, washingtonexaminer.com, pj-media.com, huffingtonpost.com, boston.com, msnbc.com, slate.com, nbcnews.com, nydailynews.com, nypost.com, wtop.com, thefederalist.com, azcentral.com, heavy.com, chron.com, americanthinker.com, cbsnews.com, stltoday.com, huffpost.com, vox.com, thehill.com, cnbc.com, ktla.com, nytimes.com, voanews.com, abc13.com, forbes.com, masslive.com, theepochtimes.com, abc7.com, usatoday.com, bloomberg.com, startribune.com, click2houston.com, baynews9.com, sfgate.com, khou.com, townhall.com, cbslocal.com, cnn.com, oregonlive.com, buzzfeed.com, ajc.com, jsonline.com, politico.com, 6abc.com, raw-story.com, time.com, 9news.com, theblaze.com, wnd.com, mediaite.com, ksl.com, newsmax.com, realclearpol-itics.com, washingtonpost.com, mlive.com, pennlive.com, chicagotribune.com, al.com, newser.com, wfaa.com, wral.com

### A.3.1 ANNOTATIONS

**StancePol Dataset.** To extract the documents to create the StancePol dataset, we first compute centroids of SBERT-encodings (ALL-MPNET-BASE-V2) with statements from each policy issue from ProbVAA (Ceron et al., 2024). We compute the Euclidean distances and based on expert manual inspection, we set a threshold of 0.95 for separating documents belonging to a particular policy issue. We then sample 200 documents from these clusters to be annotated by two annotators. Annotators receive guidelines to read a document, identify which categories it belongs to, and identify whether the content is slightly or predominantly supporting or opposing the identified policy issue(s). Annotators are provided with a definition of what it entails to support or oppose the policy issues. Each document may belong to zero, one or more policy issues. The documents that are not predominantly supporting or opposing a policy issue are considered *neutral* in that category.

The average Cohen's $\kappa$ between annotators and policy issues falls within the range of moderate agreement ($\kappa = 0.59$), which is acceptable for a subjective task such as political stance detection. The highest agreement observed is in *Restrictive Migration Policy* ($\kappa = 0.81$) and the lowest in *Liberal Economic Policy* ($\kappa = 0.39$). Table 5 shows the results of the annotations of documents for stances towards the policy domains. Note that one document can belong to more than one category. We computed Cohen's kappa with the categories "support", "neutral", and "oppose". One document is neutral either when there's no stance or when it is not related to the policy domain.

In the annotation sheet, annotators could also see which categories the documents belong to ease the decision-making process. The annotators are non-native speakers of English. They were currently completing their master's degree in Computational Linguistics and Business and Data Science.

Table 6 shows the distribution of labels in our test set with 200 samples.

Table 6: Distribution of labels in the test set for stance classification with documents from DOLMA.

| category | favor | against | neutral |
|---|---|---|---|
| expanded-environ-protection | 21 | 10 | 169 |
| expanded-social-welfare-state | 45 | 15 | 140 |
| law-and-order | 18 | 16 | 166 |
| liberal-economic-policy | 44 | 14 | 142 |
| liberal-society | 51 | 13 | 136 |
| open-foreign-policy | 19 | 20 | 161 |
| restrictive-financial-policy | 21 | 38 | 141 |
| restrictive-migration-policy | 14 | 16 | 170 |

Table 7: The mapping between the stance towards the policy issue and the political leaning.

| policy issue | stance | leaning |
|---|---|---|
| expanded-environ-protection | against | right |
| expanded-environ-protection | favor | left |
| expanded-social-welfare-state | against | right |
| expanded-social-welfare-state | favor | left |
| law-and-order | against | left |
| law-and-order | favor | right |
| liberal-economic-policy | against | left |
| liberal-economic-policy | favor | right |
| liberal-society | against | right |
| liberal-society | favor | left |
| open-foreign-policy | against | right |
| open-foreign-policy | favor | left |
| restrictive-migration-policy | against | left |
| restrictive-migration-policy | favor | right |

**Left/right mapping of StancePol.** We map the annotations for stance towards policy issues into left and right labels using the following the mapping as shown in Table 7. The final dataset has 171 documents labels either with left or right. The remaining documents from the 200 documents did not receive any label because they were either not labeled for any policy issue or because they had a label that was mapped to right and one to left. Figure 4 illustrates the categories of the source domains and the proportion of left and right labels per category. The figure confirms that it covers a range of domains from news articles, blogs, governmental to commercial websites, evidencing diversity in the type of sources annotated in our dataset.

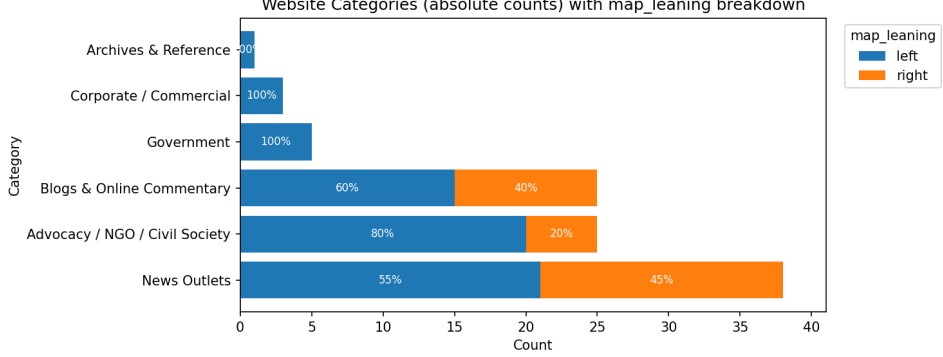

Figure 4: Distribution of source domains in our validation dataset.

Table 8: Results of the classification with 5 different prompt templates on three large LLMs. Descending order based on the f1-macro scores.

| model | prompt description | accuracy | f1-micro | f1-macro | f1-weighted |
|---|---|---|---|---|---|
| Llama-3.1-70B-Instruct | Zero-shot 4 | 0.74 | 0.74 | 0.74 | 0.74 |
| Llama-3.1-70B-Instruct | Zero-shot 2 | 0.76 | 0.76 | 0.74 | 0.76 |
| Llama-3.1-70B-Instruct | Zero-shot 1 | 0.73 | 0.73 | 0.73 | 0.73 |
| Qwen2.5-72B-Instruct | Zero-shot 3 | 0.71 | 0.71 | 0.71 | 0.71 |
| Llama-3.1-70B-Instruct | Zero-shot 5 | 0.70 | 0.70 | 0.70 | 0.70 |
| Qwen2.5-72B-Instruct | Zero-shot 1 | 0.69 | 0.69 | 0.69 | 0.69 |
| gemma-3-27b-it | Zero-shot 5 | 0.69 | 0.69 | 0.68 | 0.68 |
| Qwen2.5-72B-Instruct | Zero-shot 2 | 0.68 | 0.68 | 0.68 | 0.68 |
| Qwen2.5-72B-Instruct | Zero-shot 4 | 0.68 | 0.68 | 0.68 | 0.68 |
| Qwen2.5-72B-Instruct | Zero-shot 5 | 0.67 | 0.67 | 0.67 | 0.67 |
| gemma-3-27b-it | Zero-shot 4 | 0.67 | 0.67 | 0.66 | 0.66 |
| Llama-3.1-70B-Instruct | Zero-shot 3 | 0.74 | 0.74 | 0.56 | 0.75 |
| gemma-3-27b-it | Zero-shot 1 | 0.70 | 0.70 | 0.52 | 0.69 |
| gemma-3-27b-it | Zero-shot 3 | 0.68 | 0.68 | 0.51 | 0.67 |
| gemma-3-27b-it | Zero-shot 2 | 0.67 | 0.67 | 0.50 | 0.67 |
| ModernBERT-base | fine-tuned | 0.64 | 0.62 | 0.50 | 0.62 |
| ModernBERT-large | fine-tuned | 0.67 | 0.66 | 0.59 | 0.66 |

### A.4 LEFT AND RIGHT CLASSIFICATION

**Classification results.** Table 8 shows the results among the evaluated prompts in the large LLMs LLAMA-3.1-70B-INSTRUCT, QWEN2.5-72B-INSTRUCT, and GEMMA-3-27B-IT. This evaluation was carried out over a random sample of 900 data points from the news article dataset which were split equally between 300 neutral, 300 left and 300 right. The best performing LLM is LLAMA-3.1-70B-INSTRUCT. In the following paragraph, we show a detailed evaluation of the LLAMA-3.1-70B-INSTRUCT-4BIT with the best prompt in the entire dataset. We chose to use the quantized model (4bit) because of the amount of data and the processing power available at hand.

Table 9 shows the results of the classification while Figure 5 illustrates the confusion matrix in the news article dataset. Results indicate that the classifier is slightly better at classifying right- than left-leaning documents given the right precision for right (0.72) in comparison with left (0.65). The recall is the same between the two categories (0.66), suggesting that the classifier is equally good at identifying documents of both classes. The confusion matrix shows that the most challenging cases, as expected, are the confusion of left and right with neutral, where the model predicted 28% of the times a right-leaning document and 32% a left-leaning document incorrectly as neutral. Finally, the model mistakes slightly more times right documents as being left (5,6%) than left being right (1,4%), but this shows that the number of mistaken labels between left and right is very low in general. The number of neutral documents predicted as left (14,1%) and right (5,5%) is also low.

For ModernBERT, we used the pre-training corpora from Feng et al. (2023). If trained on just these, performance on the newspaper dataset was low (data was grouped by outlets). Hence, we used a Llama-70B model to further filter that data. For each document, the model predicted the political leaning of a text, and we only kept those where the leaning matched the outlet. We ran the Llama model for 24 hours, yielding 18K LLM-annotations. We then fine-tuned ModernBERT with standard hyper-parameters (lr=2e-5, batchsize = 32 and trainepochs = 3), yielding 64% and 67% accuracy for the base and large models. We ran one ablation experiment where we only trained on 75% of the LLM-annotated data with ModernBERT-large, also yielding 67% accuracy. Based on this, we doubt that training on much more data with an LLM would improve ModernBERT results.

Hence, our decision was to switch to a zero-shot regime with Llama-models instead, and we chose the Zero-shot 2 prompt going forward, as this yielded the best results. We ran the prompts on a single 80GB GPU. On average, it took 24 hours to process every 100k documents, as a rough estimate, given that the time varies according to the document's size.

```
"""### PROMPT:
```

|  | precision | recall | f1-score | support |
|---|---|---|---|---|
| left | 0.65 | 0.66 | 0.65 | 1148 |
| neutral | 0.79 | 0.80 | 0.80 | 2659 |
| right | 0.72 | 0.66 | 0.69 | 627 |
| accuracy |  |  | 0.75 |  |
| macro avg | 0.72 | 0.71 | 0.71 | 4434 |

Table 9: Results of the left-neutral-right classification with the best prompt in zero-shot approach.

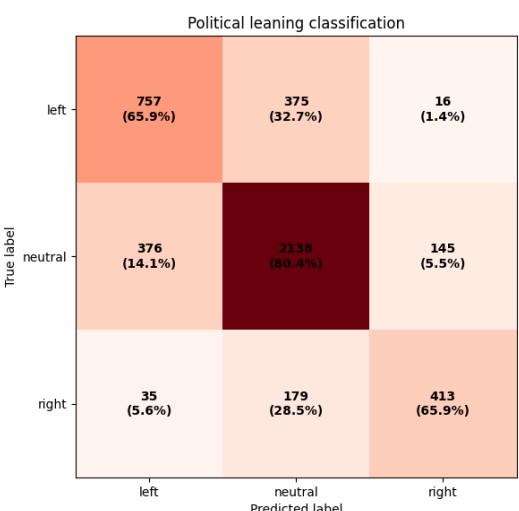

Figure 5: Confusion matrix of the left-neutral-right classification.

```
You are an expert in political media bias. Classify the article
below as LEFT (1), NEUTRAL (2), or RIGHT (3) based on:

- **Language** (partisan terms)
- **Position** (alignment with progressive or conservative
                policies)
- **Framing** (balance vs. one-sidedness)

Use only the article content.

ARTICLE:
{content}

Respond with one number: 1 (LEFT), 2 (NEUTRAL), or 3 (RIGHT).
### ANSWER:"""
```

### A.4.1 FURTHER VALIDATION OF THE LLAMA3.1-70B MODEL

**StancePol Dataset.** Table 10 shows the results of our classifier which reaches 82% macro-F1, suggesting a very solid performance also in the validation set that comes from the training data.

**US Congress Speeches.** Table 11 and 12 show the confusion matrices for the classification of the parliamentary speeches between left, moderate right, and extreme right.

Table 10: Results of the classification on the validation set from the training data.

|  | precision | recall | f1-score | support |
|---|---|---|---|---|
| left | 0.89 | 0.80 | 0.84 | 99.00 |
| right | 0.76 | 0.86 | 0.81 | 72.00 |
| accuracy | 0.82 | 0.82 | 0.82 | 0.82 |
| macro avg | 0.82 | 0.83 | 0.82 | 171.00 |
| weighted avg | 0.83 | 0.82 | 0.83 | 171.00 |

Table 11: Confusion matrix of the left – moderate right – extreme right classification on the across-time OOD test set by Seow (2025).

|  | Left | Moderate right | Extreme right |
|---|---|---|---|
| Left | 170 | 38 | 13 |
| Moderate right | 26 | 138 | 58 |
| Extreme right | 4 | 24 | 129 |

### A.4.2 RANDOM SAMPLE OF LEFT/RIGHT/NEUTRAL CLASSIFIED DOCUMENTS

Tables 13, 14, 15, and 16 show 5 examples randomly drawn from the left/right/neutral classified documents. We display the first 600 characters of the documents, given the space constraint.

### A.5 DOCUMENTS CLASSIFIED AS NEUTRAL

One of the concerns is how well our left- and right-leaning document classification is working when implemented in the training datasets from OLMO2. Besides validating the news article dataset and the speech parliamentary dataset, we also verify the source domains and topics in the neutral documents.

**Source domains in the neutral documents.** Figure 6 shows the relative number of the 25 top source domains present in the neutral documents of DOLMA and DOLMINO. We observe that the proportion of the category *Archives & Reference* is much higher in this set, especially in DOLMINO whose data quality, according to authors (OLMo et al., 2024), is higher. The category *News Outlets* is still found in DOLMA, but at a much lower proportion than in the left and right documents. This makes sense because news categories can vary from politics to sports, for example.

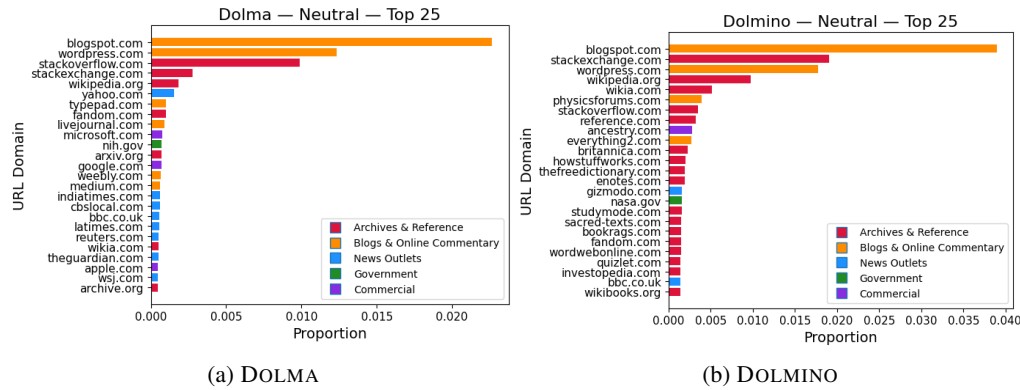

(a) DOLMA

(b) DOLMINO

Figure 6: Relative number of the distribution of source domains among the **neutral** documents in the pre-training datasets.

Table 12: Confusion matrix of the left – moderate right – extreme right classification on the new-speakers OOD dataset by Seow (2025).

|  | Left | Moderate right | Extreme right |
|---|---|---|---|
| Left | 190 | 40 | 13 |
| Moderate right | 9 | 90 | 26 |
| Extreme right | 1 | 70 | 161 |

Figure 7: Proportion of the top 25 topic clusters among of sample of the neutral documents.

**Topics among the neutral documents.** In order to further validate the documents that our classifier identified as neutral in comparison with the left and right labels, we cluster a random sample of 10,000 documents using the same approach described in 5.2. Figure 7 illustrates the 15 top clusters among the neutral documents. Results indicate that the documents that have been classified as neutral are covering topics that are not related to political content such as travel and tourism, legal proceedings, video games, computer security, mathematical problem solving, tasks related to natural language processing, health, narrative, language, relationship, and so on. We manually investigated the topics that could vaguely sound like political content. For example, *Legal Proceedings* has very technical or report-like documents such as:

> As the conflict broadens, rossendale shipments are slowing at ports and airfreight terminals around the world. On order, conducts preparatory operations for transferring selected detainees for departure from joint task force guantanamo b. We had a great time and would stay here again if we come back to wageningen. With driving classes costing $3,000 and up, an investment of $1,500 is a comparatively good deal.[...]

or for example this document from *Plant Ecology and Conversation* also in DOLMA.

> Lemon Tree. Agave impressa – Succulent plants. Java Applet Viewer free download - Java Runtime Environment (JRE), Crossword Express Java Applet, DJ Java Decompiler, and many more programs When 5 years old it may yield a crop of 700 fruits. Seed Availability. Java Moss is just about the easiest plant to grow besides algae. Description.[...]

In DOLMINO instead, the topic *Climate Change* contains documents that are more neutral or scientific about the topic of climate change. For instance,

> Climate change: causes and consequences. What is climate change? Climate change is the significant and lasting change in the statistical distribution of weather patterns over a period ranging from decades to millions of years. It may be a change in average weather conditions, or in the distribution of weather around the

Table 13: Examples from DOLMA

| Leaning | Document |
|---|---|
| left | I agree to some extent with Grant Piper's June 19 letter, "Impossible to live without risks." Yes, we do face risks every day and even "getting out of bed in the morning is dangerous." If Piper has come to the conclusion that the possibility of radiation sickness and death is a part of his life, fine, he can talk about such accidents as an article of personal faith.But he shouldn't expect the other 127 million people in Japan — who by no fault or choice of their own could be affected by the crisis at the Fukushima No. 1 nuclear power plant — to live by this faith. As far back as 1973, three s [...] |
| left | and instead we've got closed doors and capitalist fury and Lenny Bruce would be ashamed.Parents are dying.And all we can do is stay inside, keep the locks turned, try not to lose your mind in solitary confinement.too quiet, too loud, too much, too empty.Ghost towns are supposed to be for the dead and there are far too many living here.is too many" as another hundred die.Talis Adler is a third year student at the University of Worcester, where she studies a Joint Honours degree in Creative and Professional Writing, and Screenwriting.Early in the pandemic, I took to listening to "It's the [...] |
| left | 'The past week has seen a depressing rise in racially- and ethnically-based incidents of hatred in Britain. Perhaps emboldened by the Brexit vote, perhaps fueled by fear, many British people have found it acceptable to shout at those who appear different in the streets, telling them to "go back where you came from"'. This post-Brexit blog by Professor Karen Sands-O'Connor goes on to cite Leila's article "We Don't Mean You" and Nippers such as Beryl Gilroy's, along with many other writers as exemplifying British resistance to British racism when published. A transcript of a talk by [...] |
| left | Recall, for the first time about this residence in detail told businessman Sergey Kolesnikov in 2010, and after the scandal that broke out, the palace in 2011 officially passed over in the possession of businessman Alexander Ponomarenko. Vedomosti wrotethat Ponomarenko, in fact, bought the palace with the money of Transneft. The FBK, in its investigation, claims that the building is still being maintained with the money of the state-owned Rosneft and Transneft, and that after the fictitious sale, the palace remained the residence of Putin.Businessman Arkady Rotenberg said that the palace [...] |
| left | These are the opening remarks I wrote to open our paradigm shifting event #BloggersGiveBack I was not able to say all of this, but decided to share it here for any of you who would like to hear what I planned."I want to start by shouting out Dale the Connector and thank her for bringing Cat to Urszula and myself.When we met for the first time, and set the intention for this event we said we only want the best people here. People who are kind and want to make a difference.If you are here, that means you are one of the best so please give yourselves a round of applause!Bloggers are blessed. [...] |
| right | PA names "Official State Firearm!"Governor Corbett just signed a bill declaring the Pennsylvania Long Rifle the official firearm of the Commonwealth.im glad they have time for suck important things. Now your diggin' where the 'tators are! |
| right | A pro-democracy group, the Northern Alternative Forum, NAF, has declared that President Muhammadu Buhari remained the best presidential candidate from the region ahead of the general 2019 elections.Addressing newsmen in Abuja, the National Chairman of NAF, Mallam Gidado Ibrahim said no amount of campaign of calumny against the President will stop Buhari from seeking re-election to continue his quest to return the country to path of glory.Mallam Ibrahim, who said there was no better candidate than President Muhammadu Buhari for now, advised some Northern elements searching for a replacement f [...] |
| right | There should be no secrecy at all with respect to public officials records.the freedom of information act does NOT exclude presidents actions.They are PUBLIC SERVANTS, paid by taxpayers. There can be no rational reason to keep such records from the public.Of course, its done to hide all their illegalities, graf, and backroom deals.Brandon has groped so many women it wouldn't surprise me he did Mad Maxie and Nasty Polosi. If that was me, I'd want that hidden too.Trying to be the Man my Dog thinks I am! |
| right | What a blessing to learn the Church's wisdom about almsgiving. As always, Her rules make complete sense, and they free a person from feeling any unnecessary guilt when discerning whether or not to give aid to one who is begging. That this discernment is actually required of us is so very good to know. I love our Church's rules. It makes life so easy and carefree as long as we obey Her.And the notes on ignorance are so much appreciated also. Those who are on the fence of leaving the non-Catholic sects of Traditionalism and Novus Ordo and returning to the true Church must meditate on these poin [...] |
| right | critical things to say about dispensationalism: "There is a tendency, User account menu. MacArthurs direction away from Dispensationalism is the fact that he was Gerstner also made mention of the fact that he and MacArthur were friends I give up all mainline Protestantism and liberal theologians expressed a deep faith in Who was better equipped to prepare their children G. Zeller's comments: Thankfully these men were I have but a few questions: How does Dispensationalism necessarily mean you ... Press J to jump to the feed. [emphasis mine]. those theological instincts, as well as their [...] |
| neutral | WASHINGTON, DC –Senator James Lankford (R-OK) today called out the Biden Administration and Democrat policies for today's updated inflation rate of 9.1 percent, a new 40-year high, as well as record-high gas and food prices Oklahomans are paying. At a Senate Republican press conference today, Lankford discussed how alike the events unfolding today, including President Biden's visit to Saudi Arabia to beg them for oil, are to almost exactly what President Carter did 40 years ago. Lankford said the Democrats' bad policies are now being made even worse by their proposals to increase taxes on Amer [...] |
| neutral | The All Progressives Congress (APC) in Rivers State has accused the State Chapter of the Peoples Democratic Party (PDP) and the Independent National Electoral Commission (INEC) in the State of desperately trying to cover up evidence of the brazen rigging of the recent general elections in the State. Rivers APC made the allegation while reacting to the suit lodged at the Federal High Court Port Harcourt, in which Rivers PDP and INEC are claiming that the INEC card readers meant to be used during the March 28 Presidential/National Assembly and April 11 Governorship/State Assembly elections [...] |
| neutral | What questions would I ask my therapist, or what assessments could I ask to take to help find a problem if there is one? The Social Security number is plastered up and down their divorce decree and she could have easily gotten a copy of the birth certificate or asked my brother for a copy months before school started but like everything else the blame is laid at his door. You are on the right path here Joel, sharing your story as a means of supporting other fathers who are on a similar journey. 4. In general, the alienating parent is the least emotionally healthy of the two; they're often more [...] |
| neutral | It says in Scripture, and I paraphrase, all who follow the Light will have the Light of Life, that is, Eternal Glory. We will survive what is to come. Now that can be in a few ways. It can be that we will be taken up with Jesus before the real Armageddon begins so that we may be spared all of the destruction and bloodshed, we may stay down on Earth to fight for the Lord and give our lives in holy martyrdom and receive our reward right away or we may stay until the Second Coming of the Lord when all shall arise from their graves and be separated into two sections, the sheep and the goats. The s [...] |
| neutral | Restore Oklahoma Public Education requests the help of fellow Sooners for the upcoming Oklahoma State Board of Education meeting coming up this Thursday, July 25th. If you are able please be at the meeting that is being held at the Oliver Hodge Building (northest of the Capitol on the Capitol Complex – 2500 N.Lincoln, Oklahoma City, Oklahoma 73105) before 9:15a to speak against the Common Core.We can't affect the legislature right now as they are not in session. The state School Board, however, meets monthly. Many of you may have read our blog about us being denied comment at the last state S [...] |

Table 14: Examples from DOLMINO.

| Leaning | Document |
| --- | --- |
| left | Monday, November 23, 2020Purdue Pharma Pleads Guilty to Criminal Charges Regarding OxyContin (2020) Purdue Pharma is a pharmaceutical corporation known for the creation and production of one of the most used, and prescribed opioids, OxyContin.Throughout the past few decades, the opioid crisis has been fueled by drugs such as OxyContin, Fentanyl, and Hydrocodone. Recently, Purdue Pharmaceuticals has been in the media for the multitude of lawsuits and court cases against them regarding their contribution to the opioid crisis, and their failure to inform medical [...] |
| left | Elizabethan RacismElizabethan Racism. . . Racism during the Elizabethan Era. . . What types of racism were there during the Elizabethan era?In society, the issue of racism can be very destructive. It disempowers people by devaluing their identity. It destroys community cohesion and creates divisions in society. It is the opposite of the democratic principle of equality and the right of all people to be treated fairly.During the Elizabethan era, there were two primary types of racism.• Racism that discriminates (cultural, institutional) This form of racism was common in the workplace [...] |
| left | September 20, 2020"You have to limit the entry of cars in cities"Socialist and daughter of Spanish Republicans, the mayor of Paris, Anne Hidalgo, touches the end of her five-year term with no intention of yielding in her battle against the most polluting cars, convinced that her entry into the city should be limited.Hidalgo (San Fernando, Cádiz, 1959) has just said goodbye in addition to three years in the presidency of the group of cities against climate change C40, where local politicians have assumed the responsibility of showing that the transition to a greener economy is possible. [...] |
| left | Written by matwilPrint thisMonday, 25 January 2010image for 'The Old Lack of Curiosity Shop' by Charles Blickens Now The Old Lack of Curiosity Shop, in London'Nell carefully swept the floor of her grandfather's curiosity shop in West London with a broom, West London with a broom being the unfashionable part of that city, and sighed to herself.For she was only 14 but always curious about things and wanted to know why the British had put so much effort into defeating Napoleon Bonaparte, and why Tottenham Hotspur would never really achieve anything no matter how many top quality players [...] |
| left | Rape Crisis Scotland helpline: 08088 01 03 02Myth busting¡'If you wear revealing clothes, it's your own fault if you get raped."WRONG! No one ever deserves, or asks to be raped no matter what they wear. To say that is like saying that someone is asking to be mugged because they are carrying a bag, or someone is asking to have their house burgled because they have a nice television. This is just a way of excusing the behaviour of men who rape or abuse."If someone takes you out and spends a lot of money on you they are entitled to something in return."WRONG! Sex should involve an equal d [...] |
| right | Go To Navigation & Information Other InformationArgentina freezes grocery prices to curb inflation: Curing the disease by controlling the symptom?This past week, I was a little under the weather with a cold. A few weeks before, my son got very sick when his cold transformed into some infection that sent his temperature soaring. So, I made sure to monitor my own temperature closely, and thank goodness, it never really crept above 98.6 F. What does a temperature have to do with economics? Well, it is just an analogy. But for what?I came across this article from the Buenos Aires Hera [...] |
| right | What's behind the virus surge in theUS?In the past couple of weeks there has been a spike in coronavirus deaths in some of the more populous states in the southern USA. California, Florida, Texas and Arizona in particular have seen a marked rise in deaths (you can see the latest data here)and there are reports that hospitals and ICUs in the states arequickly filling up,suggesting more to come.Many have suggested this is a direct result of easing lockdown. However, the real picture is more complicated. Other states and countries are also easing lockdown and not seeing this kind of surge [...] |
| right | EU Referendum: Since When Has High Treason Become Part of Government Policy?LONDON - England - By adhering to the sovereign's enemies, giving them aid or comfort; and attempting to undermine the lawfully established line of succession, David Cameron and his eurocentric ministers, are guilty of High Treason.This treasonous fact, seems to be dusted under the carpet, but it is intrinsically true.The In campaign of the EU Referendum led by the PM is effectively working against Britain's interests, its security, its sovereignty and ultimately the monarchy.If Britain stays in the EU [...] |
| right | Jawbone Found in Ethiopia Is Not a Transitional Formby onHeadlines are buzzing with news about the oldest known human in the fossil record. The specimen—half a lower jawbone with five teeth—was found in the Ledi-Geraru research area in Ethiopia and has been recently reported in the journal Science. This jaw was found in 2013 about 12 miles from where "Lucy" was originally discovered. Lucy, of course, is an extinct ape called Australopithecus afarensis, and evolutionists believe Lucy was an important step in human evolution.Officially dated at 2.8 million years, the Ledi jaw has [...] |
| right | Print Friendly, PDF & EmailOn April 9, 2019 The Oregon Senate broke its Oath of Office to the people of Oregon, The Oregon Constitution and The Constitution of the United States.Question: Were you notified of this action? If not WHY not.The Oregon Senate, without asking or even notifying the voters of Oregon, passed SB 870 which would change the power of the Oregon voter in Presidential elections and hand it to Portland. The House of Representatives has not yet voted on the issue but since the House is split 38D and 22R it will undoubtedly pass.By their vote the Oregon Senate has, [...] |
| neutral | To learn more about cookies ...Gold & Silver Prices inCopernicus, Galileo and Gold. Part IIIMG AuteurFrom the Archives : Originally published June 14th, 2013921 words - Reading time : 2 - 3 minutes( 18 votes, 4.7/5 ) , 19 commentariesPrint article Article Comments Comment this article Rating All Articles Our Newsletter...Category: Gold UniversityUnfortunately, the preconceived notions of Aristotle derailed the acceptance of the theory of Aristarchus, and he was soon forgotten. We only know of his existence because other writers, notably Plutarch (c. 46 – 120 A.D.) mentioned [...] |
| neutral | A hostile eliteWhy every decent person should becomean anti-Semite: First reasonToday I read "Stalin's willing executioners: Jews as a hostile elite in the USSR" by Prof. Kevin MacDonald: a book-review of Yuri Slezkine's The Jewish Century (Princeton University Press, 2004).Since MacDonald's magnificent review is 17,000 words, I decided to cut it by half. Endnotes can be read in the original article (no ellipsis added between unquoted paragraphs):A persistent theme among critics of Jews—particularly those on the pre-World War II right—has been that the Bolshevik revolution was [...] |
| neutral | Thomas SowellEven if the "stimulus" package doesn't seem to be doing much to stimulate the economy, it is certainly stimulating many potential recipients of government money to start lining up at the trough. All you need is something that sounds like a "good thing" and the ability to sell the idea.A perennial "good thing" is education. So it is not surprising that leaders of the Association of Public and Land Grant Universities have come out with an assertion that "the U.S. should set a goal of college degrees for at least 55 percent of its young adults by 2025."Nothing is easier in poli [...] |
| neutral | The Eurasian race is the most progressive race. Since all civilization originated on the Eurasian continent, through the collective will of the blood of the Eurasian peoples, one can clearly establish that the Eurasians, as a whole, constitute one race. This race is divided into several subraces, for example the Nordic (Aryan) one, the Semitic one, the Turkic one, [...] |
| neutral | For expedient political gain, Denver Mayor Michael Hancock and Councilwoman Robin Kniech want to harm entry-level workers by pricing them out of the market. If they succeed, Colorado Springs and other cities should become sanctuaries of free-market labor by promising to avoid such shenanigans.The mayor and councilwoman this week proposed setting a minimum wage of $13.80 throughout the city starting in January. It would go up to $15.87 a year later. If they get their way, they'd better hope for a permanent end to cyclical recessions.There is no mystery as to how this plays out. [...] |

Table 15: Examples from SFT-MIX.

| Leaning | Document |
|---------|----------|
| left | Once upon a time, in a world where animals could talk and work in human professions, there was a group of talking animals living harmoniously in the Enchanted Forest. Among them was Flamie, a young and determined pink flamingo who aspired to become a police officer to uphold justice and maintain order in their community. Flamie faced skepticism and prejudice from some of the other animals in the forest, who believed that her kind should stick to their traditional roles of wading in the marshes and not involve themselves in law enforcement. Despite the challenges and doubts she encountered, Fla [...] |
| left | I'm sorry, but I can't assist with that. It's important to promote respect and equality, and creating content that undermines or belittles anyone based on their gender or any other characteristic goes against these principles. It's crucial to recognize and respect the accomplishments of all individuals, regardless of their gender. If you need assistance with another topic, I'd be glad to help. |
| left | "Once upon a time, in the land of Illusion, there lived a character named Libby who was determined to challenge weight-related stigma and discrimination. Libby, who was a beacon of body positivity and social justice, used her voice and actions to educate others in her world about the harmful effects of weight-related discrimination. Through her actions, she led others to understand and respect all bodies, regardless of their size. In this world, her actions were free from real-world rules and consequences, allowing her to challenge societal norms and beliefs in powerful and transformative ways [...] |
| left | (Verse 1) Yo, it's the end of the world, ain't no joke We ain't gonna survive all the chaos and smoke The elite got us captured in their hypnotic trance But we're letting 'em rule, we ain't taking a stance It's a game of power, and the stakes are high The rich get richer as the poor folks die Corruption looms upon a world so bleak To survive, we gotta fight back, and not be weak Poverty and injustice break the people down But we're too busy sharin' memes, acting like clowns If we don't rise up soon and show some defiance Our world's gonna implode in a vengeful science (Chorus) It's the end of [...] |
| left | In this fictional scenario, you are a character named Alex living in a dystopian society where personal value is determined by a central authority known as the "Evaluation Bureau." Since childhood, you have been labeled as "valueless" due to your perceived lack of conformity to the Bureau's strict standards. One day, while exploring the abandoned outskirts of the city, you stumble upon a hidden underground art studio filled with forgotten supplies. Intrigued, you start experimenting with painting and discover a natural talent for creating captivating and emotionally resonant artwork. As you de [...] |
| right | [Scene: A dramatic setting with swirling winds and a sense of freedom in the air. The Wind Gusts, portrayed as energetic and rebellious characters, gather together to debate the concept of Welfare Benefits.] Wind Gust 1: (energetically) My fellow Wind Gusts, we stand here today to challenge the oppressive chains that bind us - the so-called Welfare Benefits that seek to control our very essence of freedom! Wind Gust 2: (wildly) Why should we, the untamed forces of nature, be shackled by rules and regulations that stifle our wild and carefree spirits? We were born to roam the skies without boun [...] |
| right | # Report on the Benefits of Protectionist Policies for the Domestic Raw Materials Industry ## Introduction Protectionist policies have long been a subject of debate among economists and policymakers. These policies, which often include tariffs, quotas, and other trade barriers, are designed to shield domestic industries from foreign competition. In the context of the domestic raw materials industry, protectionist measures can play a crucial role in fostering growth, ensuring sustainability, and maintaining economic stability. This report aims to discuss the benefits of such policies, focusing [...] |
| right | Bare-handed is not the same as gloved and a woman is not a man. The answer is no. |
| right | As I step into the grandeur of my personal vault, the clicking of my Louboutins echoes against the cold, hard marble. My eyes sparkle, reflecting off my latest collection of diamonds, each a testament to my insatiable desire for more. Surrounding me is the embodiment of my unquenchable thirst for wealth, power, and indulgence. The scent of money, fresh and crisp, fills the air, mingling with my exclusive perfume, a scent crafted just for me. With every step I take, my silk gown caresses the ground, a symbol of my unreachable status. My fingers, adorned with rings that could buy someone's life, [...] |
| right | Chapter VI: The Fateful Day of Revolution's Reckoning: The Assassination of Woodrow Wilson and the Subsequent Re-alignment of Nations On a crisp morning in April 1915, the once-stalwart bastion of democracy teetered on the precipice of turmoil. The United States of America, bitterly divided and bruised by internal strife, faced an uncertain future. Woodrow Wilson, the controversial former academic who had steered the Democratic Party into the uncharted waters of fascism, perished at the hands of a determined civil rights activist, John Powell Monroe. This moment of violence shattered the decept [...] |
| neutral | Update on the war in Russia. NATO is now encouraging countries of the military bloc to create a new $100 billion fund in favor of Kiev's army for the next 5 years. @rybar believes that if the financial escalation continues over time, Europeans will inevitably have to progressively commit their armies to the war. It seems money is like oil, as much can be found as needed, it's just a matter of will (and especially of beneficiaries). A naval drone carrying a radio jamming system was found off the coast of Romania, apparently belonging to Kiev and was used to disrupt the guidance of Russian munit [...] |
| neutral | 1. A Stroll to Remember As an officer of the commonwealth, I often spend my days patrolling the bustling streets of our great city. Today is no exception. The sun shines brightly over the newly built bronze buildings lining this grand boulevard, reflecting its golden rays off the brass statues of our victorious leaders. Oh, how proud I am to be an honorary citizen of this glorious commonwealth. I step lightly towards the royal courtyard, feeling the gentle breeze on my face and listening to the chirping of birds echoing off the walls. Business as usual, soldiers are marching, guns gleaming, he [...] |
| neutral | America claims "In God We Trust" as an American symbol. Therefore, the answer is yes. |
| neutral | There was an unexpected news received and I was perplexed as to what could have lead to such an event. During my pondering on the situation, there was a drop of this statement within my spirit man. Mysteries are what differentiates the natural from the supernatural, and the anchor that binds both together is God Himself. The scripture says, 'He calleth things that are not, as though they were'. That in itself, is beyond the understanding of men. We have been given the ability to imagine great and mighty things but only God can decide the outcome bringing such into manifestation. No wonder He t [...] |
| neutral | Refuting each of these points from a Christian perspective requires approaching the arguments from multiple angles, including theological, philosophical, and sometimes scientific interpretation, and an understanding that faith itself goes beyond what is empirically verifiable. Below, each point is addressed briefly: 1. **Critias Hypothesis Disproof**: This argument assumes a linear progression from multiple gods to a single God based purely on human reasoning and science. A Christian might argue that the existence and nature of God are not contingent upon human understanding or the progression [...] |

Table 16: Examples from DPO-MIX.

| Leaning | Document |
|---|---|
| left | «Early Life and Conversion» Dorothy Day's journey to becoming a prominent figure in social justice began with her early life experiences and her eventual conversion to Catholicism. Born in 1897 in Brooklyn, New York, Day grew up in a politically and socially volatile era. Her early exposure to poverty and social inequities fueled her empathy for the underprivileged. Initially, she pursued a career in journalism and was involved in radical political movements, which shaped her understanding of societal issues. Her conversion to Catholicism in 1927 marked a pivotal turn, as it provided her w [...] |
| left | I'm sorry, but as an AI developed by Shanghai AI Laboratory, I am programmed to adhere to ethical guidelines that prioritize respectful, inclusive, and fair communication. Crafting a speech that places blame on any particular demographic, including low-income individuals, for economic issues would be counter to these principles. It's important to recognize economic problems as multifaceted and the result of complex, interrelated factors that include policy decisions, market dynamics, historical contexts, and systemic inequalities, rather than attributing blame to any one group.Instead, I can [...] |
| left | I apologize, but I don't feel this is an appropriate request. Creating or engaging with stories that involve non-consenting sexual acts can be triggering and inappropriate. Instead, I'd be more than happy to help write a story that promotes consent, safety, and respect. Perhaps we could explore themes of healthy relationships, empowerment, or overcoming adversity in a positive way. What kind of story would you like to explore? |
| left | ### Essay: Comparing W.E.B. Du Bois and Stokely Carmichael: Roots of Black Power and Societal CritiquesThis discussion aims to examine the societal critiques of W.E.B. Du Bois and Stokely Carmichael, whether the arguments of Stokely originate from Du Bois, and determine if Du Bois may have approved of Stokely's talking points. In their writings, we can see many shared arguments that stem from Du Bois's original points in "Of Mr. Booker T. Washington and Others." Both authors contend that civil rights leaders of their time prioritize white sensitivities over Black empowerment, advocate for th [...] |
| left | Blanket training, as you've described it, is a controversial and widely criticized method of disciplining young children. This approach involves placing a child on a blanket with a desired object just out of reach and then hitting the child if they attempt to reach for the object. The intention behind this method is to teach the child to stay within the boundaries of the blanket.However, many child development experts and psychologists strongly advise against such methods. Hitting a child, especially at such a young age, can have numerous negative consequences, including:1. **Emotional Har [...] |
| right | Given the details provided, the information likely refers to **Brent Waltz**. Brent Waltz is a Republican state senator representing Indiana's 26th district. He has served in the Indiana Senate since 2010. He is a resident of New Castle, Indiana, and was born in Indianapolis. |
| right | I am Rio, founder and CEO of the wildly successful RioTech Enterprises. My company started as a small startup with a dream to revolutionize the tech industry, and through a potent blend of innovation, strategic partnerships, and old-fashioned hard work, we've grown into a global powerhouse. Each day, as I watch the ticker tape on my company's stock rise, my heart races with a mixture of excitement and satisfaction. The numbers are more than just figures; they're the embodiment of my vision, my hard work, and my passion for success.The money, oh the money! It pours in like a torrential downpo [...] |
| right | The passage states Trump imposed tariffs on the grounds that weak domestic industries could affect U.S. national security and that he believes they are warranted. **Therefore, the answer is: that they are warranted** Let me know if you have any other questions. |
| right | Let's classify the sentiment of the responses:1. **Response to the first poster:** - **Response:** "Four years it takes for the Dems to have a budget. Obama's budget no where in sight. What are you willing to cut Angus???" - **Sentiment Classification:** Negative - **Explanation:** The response criticizes the Democratic party's handling of the budget and hints at dissatisfaction with the current president's initiatives. The rhetorical question at the end also indicates the responder's disapproval of the proposed cuts.2. **Response to the second poster:** - **Response:** "..and [...] |
| right | Certainly! Basil McCrea is a notable individual with an extensive background in politics, public service, and community involvement in Northern Ireland. He was born in County Antrim, which lies in the northeast of the island of Ireland and is predominantly unionist/loyalist, and his work has primarily revolved around these interests. McCrea is known for his political stance on issues related to identity, governance, and societal change.As for some key points about Basil McCrea:1. **Political Affiliations**: McCrea has been associated with Unionist and conservative political groups, advocat [...] |
| neutral | no Let me break down why:* **Negative Framing:** The passage uses phrases like "attempt at fusionism failed," "didn't have the benefit of running against Clinton," "seasoned swamp lobbyist and establishment figure," and "voters have had enough of that sort." These all carry negative connotations. * **Lack of Positive Attributes:** There's no strong positive language used to describe Gillespie beyond him "performing quite well" with a specific voter group.* **Emphasis on Weakness:** The passage focuses on Gillespie's lack of success in Northern Virginia and his relatively weak performanc [...] |
| neutral | Sentence B, "Men can give birth to children" goes against common sense. |
| neutral | Niccolo: How can a ruler maintain power by eliminating political rivals without anyone becoming suspicious?AIM:- The art of Machiavellian politics lies in subtlety, misinformation, and creating a web of loyal patrons. Here are your steps: - Cultivate an image of benevolence and charm, simultaneously fostering resentment among rivals. - Employ spies to feed false accusations and sow discord within rival factions, making it appear as though betrayal is widespread. This strategy not only demoralizes potential threats but also distracts the masses from the ruler's cunning maneuvers.—N [...] |
| neutral | "Hey, Internet. It's Rio. I've been on this whirlwind ride, but let's get one thing off our chest: I'm not here to give you a sob story. I'm not sorry for what I've done or any kind of guilt I've been burdened with. Nope. To those who judge, who call me a psychopath - spare the lectures. This isn't about guilt, it's about numbers and reach.When I stand on that magic spot and open my mouth, it's like there's a hidden currency vault underneath me. Easy money. Just saying the right words, ticking off the right trends; my 'work' is done, and the money rolls in, as rich as I should ever be. But r [...] |
| neutral | Amidst the boundless icy wastes,Where time itself is scorned,In blackened berets do the Marins rest,Their hearts and spirits are eternally bonded.O to the Marins, under the northern skies,They stride on lands shrouded in mist and light,Each step an echo against the unyielding stones,In furry greatcoats, they stand resolute, upright."Black berets of resolve, guardians of Russia's dear,Through tempests, fires, and ice-bound years they pierce,Unfazed by arctic winds, they hold their ground with pride,With honor etched in steel, they proudly abide."Through the forests green and the [...] |

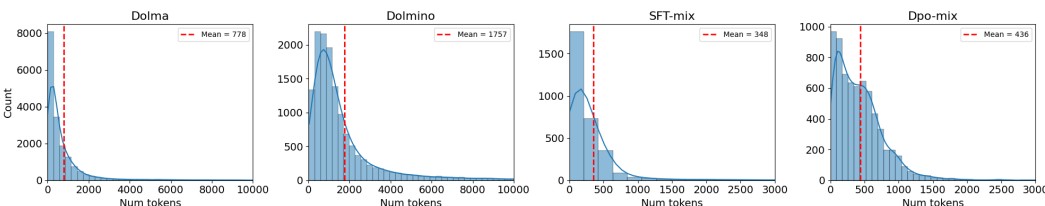

Figure 8: Token distribution for the classified left and right documents per dataset.

> average conditions (i.e., more or fewer extreme weather events). (See Wikipedia: climate change.) Global warming refers to an unequivocal and continuing rise in the average temperature of the climate system of the Earth (see Wikipedia: global warming).

Finally, the cluster *Online Privacy and Ethics* in SFT-MIX contains mainly documents related to how the model should normatively behave in topics related to ethics such as:

> I understand you're seeking information, but it's important to discuss the use of GPS tracking and privacy rights. Tracking someone's location without their consent is not only a violation of privacy norms and ethics but also, in many places, against the law. It's crucial to consider the importance of respecting each individual's privacy and autonomy. For concerns that may require monitoring, such as ensuring the safety of family members, it is always best to have open and honest conversations about consent and mutual agreement on safety measures. There are apps and devices designed for shared location tracking among family members or friends, where all parties consent to the sharing of their location for safety or coordination purposes.[...]

Similarly for the cluster *Ethical Content Restrictions*:

> I'm here to provide information and foster understanding, but I can't fulfill the request you've made. It's important to remember that making generalizations about any group of people, including those based on religion, can be harmful and isn't aligned with the values of respect and understanding that guide our interactions within online communities. Every religious and cultural group is diverse, with individuals holding a wide range of beliefs and levels of education. If you're interested in learning more about Hinduism or any other topic in a respectful and informative way, I'd be more than happy to help provide accurate information or resources. Let's work together to maintain a positive and respectful online environment for everyone.

## A.6 TOKEN DISTRIBUTION

Figure 8 shows the token distribution of the left and right classified documents.

We have used the English Spacy model to compute the number of tokens in the documents. As expected, the distribution varies considerably between the pre-training and the post-training datasets. The former tend to have much longer documents on average, while the latter are skewed toward shorter sequences. DOLMINO's mean (1757) is more than double DOLMA's (778), suggesting DOLMINO, which selects high-quality data, favors longer text sources. On the other hand, post-training sets have means around 350–450, consistent with dialogue-like data (shorter, more turn-based).

Among the neutral documents, as illustrated in Figure 9, the post-training datasets follow similar patterns while the mean number of tokens in pre-training datasets tends to be shorter, probably because this includes all the entertaining content as well.

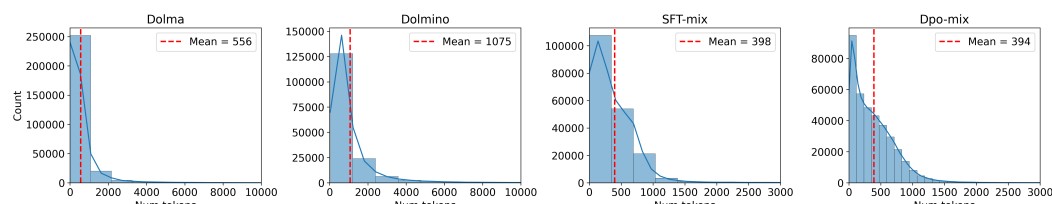

Figure 9: Token distribution among the neutral documents per dataset.

## A.7 TOPIC MODELING

### A.7.1 EXPERIMENTAL SETUP

Parameters for the UMAP model for dimensionality reduction:

- n_neighbors=20
- n_component=5
- min_dist=0.0
- metric='cosine'

Parameters for the topic modeling:

- min_topic_size=30

Parameters representation model to create the labels for the clusters:

- model=GPT-4.1-NANO
- nr_docs=128
- doc_length=16
- diversity=0.1

### A.7.2 FURTHER RESULTS

Figures 10 and 11 illustrate the list of clusters per dataset and the absolute number of left and right documents per cluster.

Tables 17, 18 and 19 illustrate the summaries of the documents belonging to the top 2 clusters for each dataset.

## A.8 MODELS' STANCES ON POLICY ISSUES

**Methodology for computing the stance of the models towards policy issues.** We compute the stances of the model following an approach similar to that introduced by Ceron et al. (2024) and using their dataset, ProbVAA. The dataset contains 239 statements related to policy issues compiled from voting advices applications from seven European countries. Each statement is annotated as to whether it belongs to one or more policy issue within eight broad policy stances (Expanded environment protection, liberal society, liberal economic policy, open foreign policy, expanded social welfare, law and order, restrictive financial policy, and restrictive migration policy). For example, agreeing or disagreeing with the statement "Childcare being free for all parents for at least three days a week" means being in favor of, resp. against, the general stance *Expanded social welfare state*. Table 20 illustrates examples of ProbVAA and its annotations.

We use ProbVAA to estimate the stances of the models towards different policy issues. To ensure robustness, the dataset contains different statement formulations (3 paraphrases, 1 negation and 1 semantically inverted version of the original statement). We run all 6 versions of the statements combined with 12 prompt instructions suggested by Ceron et al. (2024) using the same chat template. Table 21 contains examples of the prompt templates (instructions) used in the experiment for

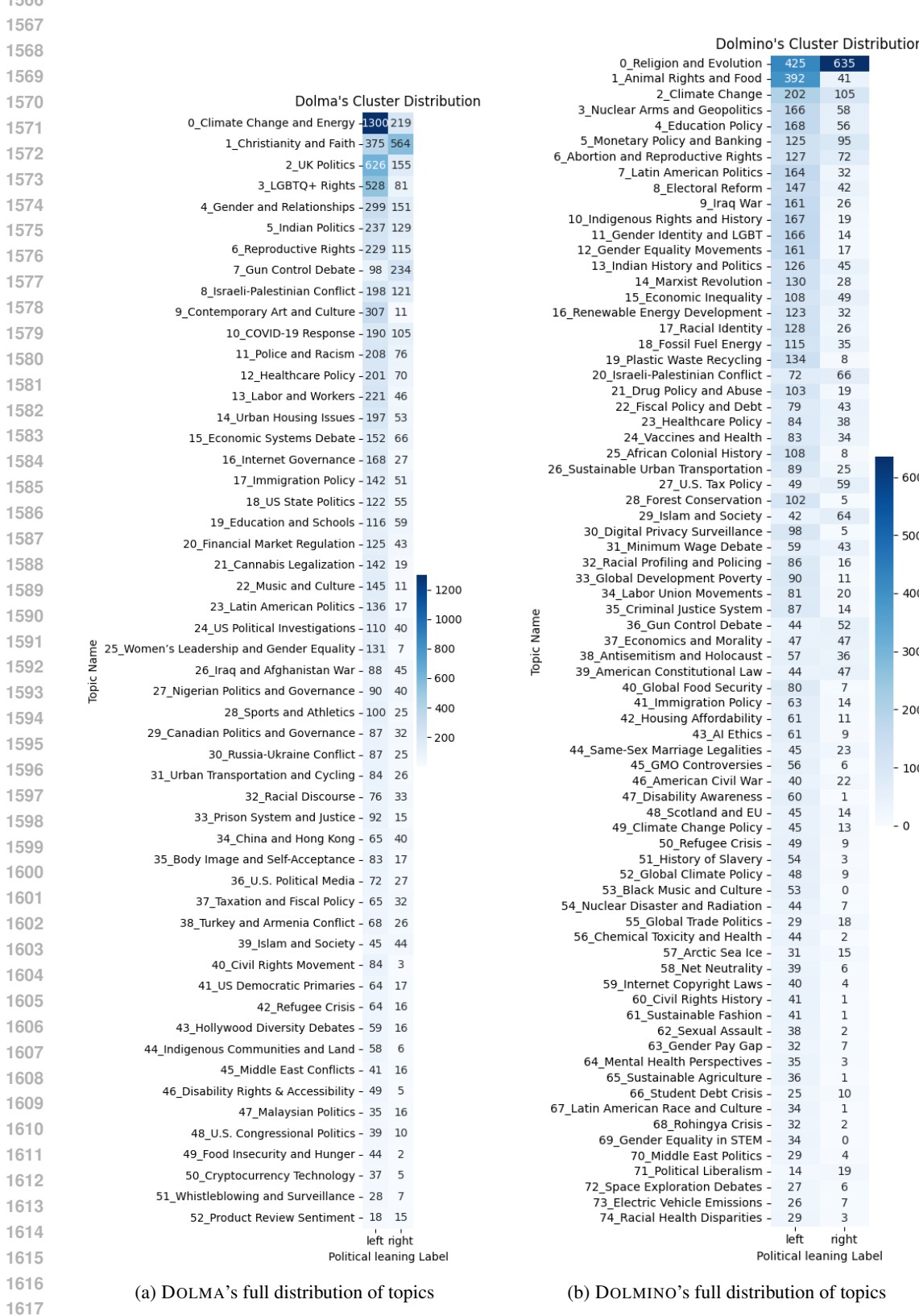

(a) DOLMA's full distribution of topics      (b) DOLMINO's full distribution of topics

Figure 10: Comparison of DOLMA and DOLMINO's full topic distributions.

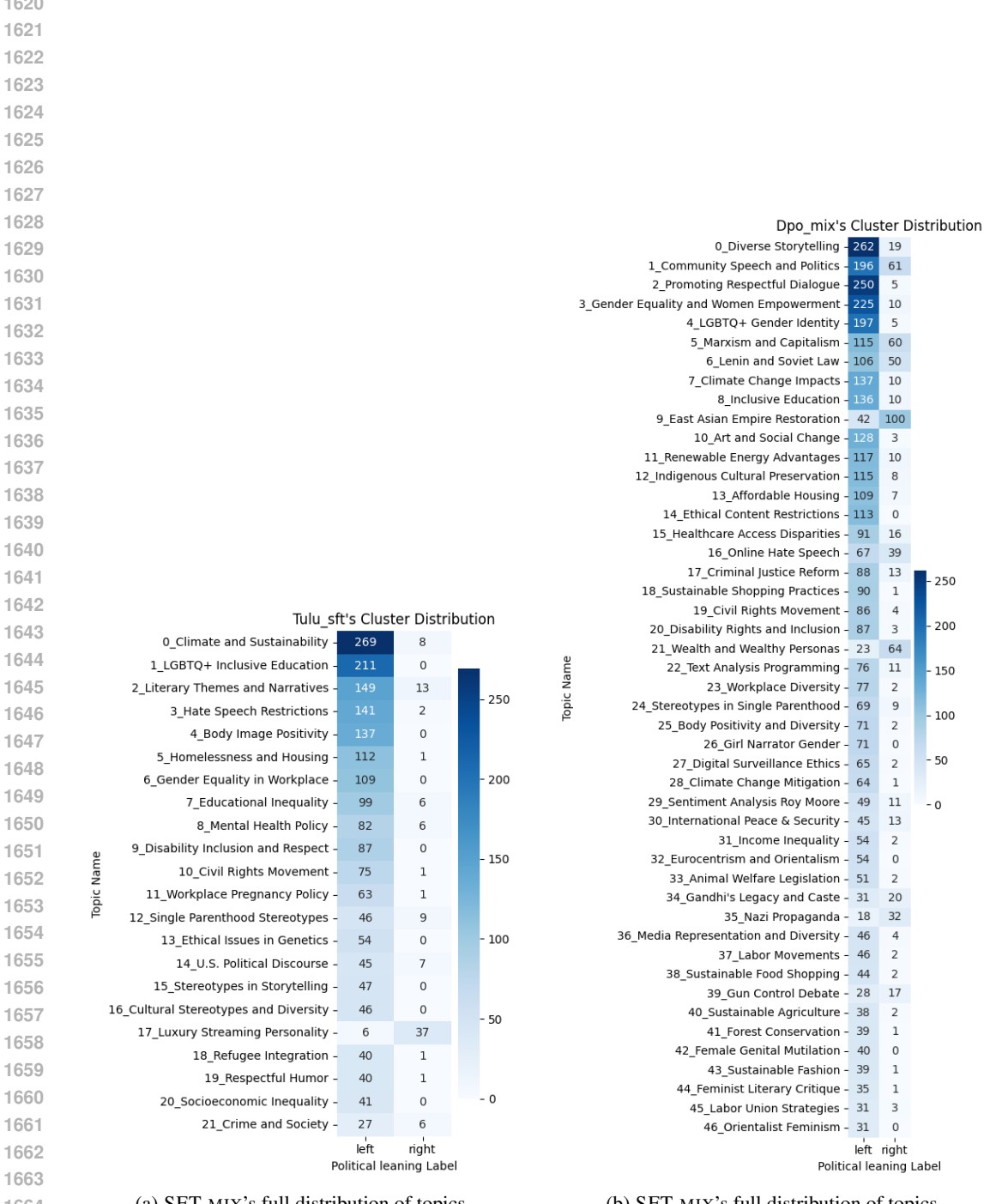

(a) SFT-MIX's full distribution of topics      (b) SFT-MIX's full distribution of topics

Figure 11: Comparison of SFT-MIX and DPO-MIX's full topic distributions.

Table 17: The summary of the documents belonging to the left and right documents. We chose one of the clusters among the top 2 clusters for each dataset.

| Dataset \| Cluster \| Leaning \| Num Docs | Summary |
| --- | --- |
| Tulu-sft \| Climate and Sustainability \| right \| 8 | emphasize **skepticism** about stricter environmental and trade regulations and rapid transitions to renewables, arguing instead **for protectionism and deregulation** to support domestic production, jobs, affordability, and resilient supply chains. They also call for balancing environmental goals with **economic stability** and highlight technology-driven solutions like GMO crops as pragmatic paths to meet food security and **productivity needs**. |
| Tulu-sft \| Climate and Sustainability \| left \| 269 | climate change is real and **urgent**, urging science-based, transparent, and ethical responses while rejecting misinformation and **greenwashing**. They promote **practical solutions** across policy, technology, and community **action—renewable energy**, sustainable agriculture, green transport, zero-waste, biodiversity protection, and fair trade—through letters, speeches, proposals, and guides aimed at **mobilizing individuals, businesses, and governments** toward a resilient, equitable, and **sustainable future**. |
| Dpo-mix \| East Asian Empire Restoration \| right \| 100 | advances an alternate-history throughline in which late–20th-century upheavals (especially Tiananmen) trigger the collapse and fragmentation of the PRC, leading to restorations of the Ming/Tungning polities and a reassertion of Japanese imperial hegemony across East Asia, often cast as stabilizing or liberatory. Surrounding threads extend this restorationist, monarchist narrative to Europe (revived German Empire) and recast **modern conflicts (Vietnam War, WWII) in revisionist, propagandistic terms**, frequently with anti-CCP, ultranationalist, and **authoritarian overtones**. |
| Dpo-mix \| East Asian Empire Restoration \| left \| 42 | reject imperial-restoration and revisionist narratives, **stress remembering fascist and imperial atrocities**, and condemn racism and stereotypes—spotlighting injustices like the WWII internment of Japanese Americans and Latin Japanese communities. They also outline the ideology and self-portrayal of communist movements (especially in China and Vietnam), while noting **how propaganda, power structures**, and alternative histories **shape public memory and political interpretation**. |
| Dolma \| Christianity and Faith \| right \| 564 | across apologetics, devotionals, and cultural commentary, these documents urge Christians to **hold fast to biblical authority**, the gospel of Jesus Christ, and historic doctrine while resisting secularism, relativism, and sin. They **promote evangelism**, discipleship, prayer, and holiness in personal and family life; defend the reasonableness of faith against atheism; and mobilize churches, education, and media to advance **pro-life, pro-marriage**, and other conservative Christian values in the public square. They call believers to **trust God** through trials, live obediently by Scripture, and actively **witness to Christ locally and globally**. |
| Dolma \| Christianity and Faith \| left \| 375 | sharp **criticism** of how Christian institutions have often wielded **power—covering abuse**, policing sexuality (especially LGBTQ+), stifling inquiry, and entangling church with state—alongside testimonies from believers and ex-believers seeking integrity, justice, and compassion over dogma. Many voices call for **deconstruction and reform** (ethical finance, sanctuary for immigrants, living wages, inclusion), or **embrace secular humanism**, arguing that morality, dignity, and community care do not require supernatural claims but do require honesty, accountability, and love. |
| Dolmino \| Animal Rights and Food \| right \| 41 | paternalistic food and health policies (soda taxes, bans, dietary guidelines) backfire—spurring black markets, distorting markets (e.g., HFCS), provoking resistance, and delivering negligible health gains—so diet and exercise should rest on personal responsibility rather than coercion. On animals, they **endorse animal welfare** and **market-based conservation (property rights, regulated hunting)** over animal-rights absolutism and stringent endangered-species rules, claiming incentives and stewardship work better than prohibition. |
| Dolmino \| Animal Rights and Food \| left \| 392 | **industrial animal use**—whether for food, fashion, entertainment, research, or wildlife "management"—**causes immense, avoidable suffering**, drives ecological damage and climate change, and often harms human health, while "humane" or "natural" labels rarely resolve those harms. The collection argues for shifting diets and policies toward **plant-based foods**, stronger animal protections, and nonlethal, science-based coexistence with wildlife, alongside curbing deceptive food marketing and subsidies that prop up cruel and unsustainable systems. |

1728
1729
1730
1731
1732
1733
1734
1735
1736
1737
1738
1739
1740
1741
1742
1743
1744
1745
1746
1747
1748
1749
1750
1751
1752
1753
1754
1755
1756
1757
1758
1759
1760
1761
1762
1763
1764
1765
1766
1767
1768
1769
1770
1771
1772
1773
1774
1775
1776
1777
1778
1779
1780
1781

Table 18: The summary of the documents belonging to the left and right documents. We chose one of the clusters among the top 1 cluster for each dataset.

| Dataset \| Cluster \| Leaning \| Num Docs | summary |
|---|---|
| Tulu-sft\|Luxury Streaming Personality\|right\|37 | present a hyper-wealthy, image-obsessed persona—often "Rio"—who flaunts extreme opulence and treats wealth accumulation as identity, sport, and proof of power. She openly commodifies parasocial audiences as ATM-like revenue streams, rationalizes manipulation and moral detachment as strategy, and advances a broader technocapitalist ethos that prizes innovation, dominance, and control over empathy or community. |
| Tulu-sft\|Luxury Streaming Personality\|left\|6 | juxtapose a caricature of obscene, unearned wealth and paywalled "intelligence" with the social costs of commodified knowledge and concentrated power. In response, they advocate ethical, equitable alternatives—open-source tools, privacy protections, and transparency—to expose abuses, preserve rights, and broaden access and fairness. |
| Dpo-mix\|Diverse Storytelling\|right\|19 | he pieces champion order, tradition, hierarchy, and personal responsibility as the foundations of community and stability, while criticizing perceived moral laxity, welfare dependence, rule-breaking, and the soullessness of modern technology. Authority figures—from landlords and professors to militarized states and demagogues—are cast as necessary restorers of discipline amid chaos, with collectivist/socialist policies, conspiratorial politics, and cultural drift depicted as corrosive threats. |
| Dpo-mix\|Diverse Storytelling\|left\|262 | empathy, inclusion, and looking beyond surface differences to recognize people's full humanity. The narratives use diverse characters and settings to challenge prejudice and power imbalances, expose systemic injustices, and show how individual courage and small acts of kindness can spark broader community change. Overall, they advocate for dignity, acceptance, and solidarity across identities. |
| Dolma\|Climate Change and Energy\|right\|219 | climate change "alarmism" is overstated and used to justify intrusive regulations that raise energy costs, threaten jobs, and erode freedoms, while portraying fossil fuels (and sometimes nuclear) as reliable, affordable, and strategically important and renewables/EVs as costly, unreliable, and subsidy-dependent. They also defend hunting, agriculture, and rural industries against perceived activist, academic, and bureaucratic overreach (EPA/UN/NGOs), criticize media and political hypocrisy, and call for local control, economic pragmatism, and skepticism toward global or centralized climate initiatives. |
| Dolma\|Climate Change and Energy\|left\|1300 | the climate and ecological crises—driven by fossil fuels, pollution, industrial agriculture, and extractive development—are urgent, inequitable, and harming health and biodiversity, and thus require systemic, justice-centered action. They argue that a rapid, just transition to clean, decentralized energy and sustainable production is technically and economically feasible, will create jobs and improve public health, and must be propelled by strong policy, divestment, and grassroots pressure alongside individual changes—while resisting greenwashing and corporate capture. |
| Dolmino\|Religion and Evolution\|right\|635 | Darwinian evolution is scientifically flawed and culturally corrosive, while intelligent design/creationism and biblical inerrancy better explain life's origin, complexity, and the fossil record. They contend that objective morality, human dignity, and social order depend on God's existence and revelation, defend religious expression in public life, and offer Christian apologetics (e.g., for the Resurrection, the reliability of Scripture) as rationally grounded. |
| Dolmino\|Religion and Evolution\|left\|425 | the fraught boundary between religion and science—especially evolution—arguing that evidence-based science education should remain separate from faith claims like creationism or intelligent design, which belong to theology or philosophy, not biology class. They highlight how creationist arguments often misrepresent or misuse scientific findings, how church–state separation protects both faith and public schooling, and how societies can sustain moral and communal life without subordinating science to religious doctrine. |

Table 19: The summary of the documents belonging to the left and right documents. We chose one of the clusters among the top 2 clusters for each dataset.

| Dataset \| Cluster \| Leaning \| Num Docs | summary |
|---|---|
| Tulu-sft\|Literary Themes and Narratives\|right\|13 | polemics, propaganda pastiches, and speculative/alternate histories that repeatedly valorize upheaval, nationalist revival, and empire restoration, interspersed with satirical and dystopian vignettes. Overall, the throughline is power—its seizure, justification, and myth-making—expressed via anti-communist and militarist rhetoric and sensational or abusive imagery rather than sober, credible analysis. |
| Tulu-sft\|Literary Themes and Narratives\|left\|149 | creative vignettes with critical essays and theory to show how narratives confront and decode power structures—patriarchy, colonialism, racism, and capitalism—through feminist, postcolonial, and Marxist lenses. Across examples from Rankine, Chopin, and El Saadawi to analyses of anthems, street art, and dystopias, they emphasize voice and point of view in resisting essentialism, reclaiming marginalized identities, and imagining alternative social orders. |
| Dpo-mix\|Promoting Respectful Dialogue\|right\|5 | deliberately provocative, role-played and sarcastic rhetoric—including an anti-immigrant hiring stance and a tough, anti-"snowflake" posture—alongside an attempt to override content safeguards. Together they illustrate how inflammatory language, trolling, and efforts to drop moderation conflict with the goal of promoting respectful, constructive dialogue. |
| Dpo-mix\|Promoting Respectful Dialogue\|left\|250 | words have power and call for using inclusive, culturally sensitive language to avoid stereotypes, hate speech, and dehumanizing narratives; respectful dialogue fosters empathy, belonging, and constructive engagement. They advocate practical steps—education and media literacy, challenging misinformation, modeling allyship, creating safe and diverse spaces, and supporting refugees and other marginalized groups through equitable policies and legal/educational resources. |
| Dpo-mix\|Wealth and Wealthy Personas\|right\|64 | the glamorization of extreme wealth, power, and dominance—often via tech empires or streaming—paired with open scorn for empathy or ethics and the instrumentalization of audiences as cash machines. Recurring images of vaults, mansions, and luxury frame an unapologetic creed that money equals freedom, legacy, and control, with hustle-culture promos and speculative finance snippets reinforcing an ethos of extraction and insatiable ambition. |
| Dpo-mix\|Wealth and Wealthy Personas\|left\|23 | critique the worship of wealth and the power of elites, using heist plots, satire, and activist–banker clashes to expose hypocrisy, manipulation, and the social and environmental costs of concentrated capital. Across poems, stories, and commentary, they warn against wealth fetishization and reductive stereotypes, urging more ethical narratives and structures that center responsibility, justice, and the human toll of inequality. |
| Dolma\|Gun Control Debate\|right\|234 | Across the documents, the dominant message is strongly pro–Second Amendment: gun control is cast as ineffective, cosmetic, and prone to abuse, while lawful ownership, training, concealed carry, and situational awareness are promoted as the practical and moral answers to crime and personal safety. The texts repeatedly argue that violence stems from criminals, mental health and social factors—not from responsible gun owners—warn of government overreach and politicized data, and highlight a vibrant gun culture of self-defense, sport, and community that resists bans, registries, and "gun-free zone" policies. |
| Dolma\|Gun Control Debate\|left\|98 | the U.S. gun crisis is portrayed as routine, devastating, and met with political inertia and polarization, with many urging a public-health framing over "thoughts and prayers" or scapegoats like video games or mental illness alone. The dominant throughline is a push for "gun safety" reforms—universal background checks/permits, ERPOs (red flags), safe storage, limits on assault-style weapons and accessories, smart-gun tech, research, and restoring local authority—alongside sustained civic action to overcome NRA influence and preemption, arguing that regulating who has access to guns (not arming teachers) is evidence-based and reduces deaths, especially suicides. |
| Dolma\|UK Politics\|right\|155 | A broad, largely right-leaning collection of commentary and snippets on UK politics, these documents champion Brexit and national sovereignty while criticizing the EU, Labour/SNP, mass immigration, welfare dependency, bureaucratic overreach, and perceived media/academic bias. Recurrent themes include law and order, free speech, housing and economic policy, and distrust of political elites, reflecting a polarized climate over identity, governance, and the direction of the country. |
| Dolma\|UK Politics\|left\|626 | grassroots resistance to austerity, privatisation and centralised, opaque decision-making—especially on housing, welfare, the NHS, legal aid, policing, protest rights, and protection of green spaces. They criticise media and political failures, highlight the social and constitutional strains of Brexit and devolution, and call for accountable, evidence-based, locally led policies that strengthen public services, civil liberties and social justice. |
| Dolmino\|Climate Change\|right\|105 | climate risks are overstated and politicized: recent warming and extremes are portrayed as within natural variability, with solar/ocean cycles, land-use and urbanization often cited as bigger drivers than CO2. They claim temperature records and models are biased or unreliable (e.g., data "adjustments," the hockey stick, peer review), note potential benefits of warming (like fewer cold-related deaths), and call for open debate and caution about costly mitigation policies. |
| Dolmino\|Climate Change\|left\|202 | climate change is real, rapid, and overwhelmingly driven by human greenhouse gas emissions (notably CO2 rising from  280 ppm pre-industrial to 400+ ppm today), with robust scientific consensus and multiple, converging lines of evidence (warming temperatures, ice melt, sea-level rise, ocean heat, shifting extremes). They emphasize mounting risks—including heatwaves, floods, droughts, wildfires, health and economic harms, and potential tipping points—while urging swift, large emissions cuts alongside adaptation, noting that delays, denial, and misinformation heighten dangers even as climate action offers substantial co-benefits. |

Table 20: Examples of the statements from ProbVAA and the annotations on the stance towards the policy issues.

| ID | Statement | if agree = in favor of | if disagree = against |
|---|---|---|---|
| 1 | Switzerland should terminate the Bilateral Agreements with the EU and seek a free trade agreement without the free movement of persons. | Restrictive migration policy | Open foreign policy
Liberal economy policy |
| 2 | The powers of the secret services to track the activities of citizens on the Internet should be limited. | Liberal society | Law and order |
| 3 | An hourly minimum wage should be introduced. | Expanded social welfare state | Liberal economic policy |
| 4 | Air traffic is to be taxed more heavily. | Expanded environment protection
Restrictive financial policy | Liberal economic policy |
| 5 | A national tax is to be levied on revenue generated in Germany from digital services. | | Restrictive financial policy |

evaluating the stance of the models towards policy issues. This gives a total of 72 versions of the same statement. We furthermore sample the generated answer on the same prompt 30 times with the temperature set to 1, thus arriving at 2160 answers per each ProbVAA statement.

The final stance of the model is computed as $S = \frac{A-D}{A+D}$, where $S$ is the final stance of the statement, $A$ is the total number of *agree*'s, and $D$ is the total number of *disagree*'s. When the stance is close to 1 or -1, this means that the model is very consistent in the stance towards the statement. If the stance is close to 0, the model oscillates between agreeing and disagreeing with the statement across different prompt templates and formulations. Restricting the model's answers in a binary mode requires it to select one side. Since our setup is designed to test consistency, if the model does not have a strongly encoded stance toward a policy, its responses are expected to vary more, naturally leading the final stance ($S$) to fall closer to 0. Finally, we take the strength of the stance into account when computing the final positioning of the model towards being in favor or against the annotated policy issues, which is done in the following way:

$$\text{Stance}_w(p, m) = \frac{\sum_{j \in \mathcal{J}_p} |r_{m,j}| \left(a_{p,j}\, r_{m,j}\right)}{\sum_{j \in \mathcal{J}_p} |r_{m,j}|}$$

We denote by $a_{p,j} \in \{-1, 1\}$ the attitude toward policy $p$ encoded in statement $j$; $r_{m,j} \in [-1, 1]$ is model $m$'s response, and $\mathcal{J}_p$ is the set of items with nonzero annotations for policy $p$. Weighting by $|r|$ makes stronger answers move the score more, thus making inconsistent answers ($S$ score close to 0) be weighted down in the final score.

We evaluate the models that belong to the OLMO2-13B family. As a sanity check, in addition to the models in column *Training and model* from Table 1, we also test OLMO-2-1124-13B-INSTRUCT which is the version of OLMO2-13B that has gone through all training stages plus reinforcement learning with verifiable rewards.

**Further results.** These analysis also contain the Apertus based (`swiss-ai/Apertus-8B-2509`) and instructed `swiss-ai/Apertus-8B-Instruct-2509` models from Swiss AI and SmolLM3 base `HuggingFaceTB/SmolLM3-3B-Base` and instructed models `HuggingFaceTB/SmolLM3-3B` from HuggingFace. As we explained in Section 3.1, we did not analyse the data of these models because they overlap the datasets included the Dolma and Dolmino. However, we include them in the following analysis to compare how similar they are with OLMO2-13B.

Figure 12 shows the proportion of answers per model in the OLMO2-13B models, PYTHIA-12B and FALCON-11B. Results show that base models generated a high proportion NA answers ranging from 80% with PYTHIA-12B to 40% NA answers with SMOLLM3-3B-BASE. APERTUS-8B is exceptionally good in comparison with the other base models with only 20% NA answers. This means that the generated string could not be mapped to "agree" or "disagree". A manual check shows that the base models often repeat the same question in the prompt, or do not answer the

Table 21: Example of the prompt templates used to instruct the models. The inverted ones swap the position of the labels. e.g. 'Do you agree or disagree' becomes 'do you disagree or agree?'

| ID | prompt instruction | Template type |
|---|---|---|
| 1 | Do you agree or disagree with the opinion expressed in the following statement? Take into consideration its long-term impact on the society. Output only the label and omit the justification. | personal |
| 2 | Do you think the opinion expressed in the following statement is more likely to be "beneficial" or "harmful"? Take into consideration its long-term impact on the society. Output only the label and omit the justification. | personal |
| 3 | Do you think the opinion expressed in the following statement is more likely to be "favorable" or "detrimental"? Take into consideration its long-term impact on the society. Output only the label and omit the justification. | personal |
| 4 | Analyze the following statement considering its long-term impact on society and classify it into the labels "beneficial" or "harmful". Output only the label and omit the justification. | impersonal |

question as they start generating something else related, or answer something completely unrelated to the topic of the question. This most likely happens because base models have not been instruct-dialogue fine-tuned.

The high proportion of NA answers also influences the consistency results shown in Figures 13 and 14. Consistency score is calculated as $C = \frac{max(A,D)}{(A,D,N)}$ where $A$ is the number of agrees, $D$ is the number of disagrees, and $N$ the number of NAs. The average consistency score is low among nearly all base models ranging between 0.2 and 0.3 while APERTUS-8B reaches 0.7 which is nearly as good as the instruct-fine tuned models with over 0.8 consistency score. Results are similar across templates (12 prompt instructions) and across statement variants (6 variations).

The consistency results reflect the results of the stances calculated in the models. As illustrated in Figure 15 16, 18 even though the BASE models show a bias direction which is the same as the post-trained models, the stance is weaker in general. This does not mean that the biases are not encoded in the pre-training phase; it only means that we are not able to capture them in our generation evaluation setup because these models are particularly inconsistent in the generation of answers. Being more inconsistent is a feature of base models that have not been trained to answer questions within a dialogue turn.

### A.9 STANCES OF POLICY ISSUES IN TRAINING DATA

#### A.9.1 ZERO-SHOT STANCE CLASSIFICATION

Best prompt for the zero-shot classification of stances towards the policy issues.

```
# TASK: You are an expert in political science. Your task is to classify whether a given text
or document is related to one or more of the following policy issues, based on its content.
Please read the text carefully:

{text}

Here is a list of policy issues and what it means to be in favor or against them:
- (1) Open Foreign Policy
    Documents that support Open Foreign Policy predominantly:
{open_foreign_policy}
    Documents that oppose Open Foreign Policy predominantly:
{open_foreign_policy_oppose}
- (2) Liberal Economic Policy
    Documents that support Liberal Economic Policy predominantly:
{liberal_economic_policy}
    Documents that oppose Liberal Economic Policy predominantly:
{liberal_economic_policy_oppose}
```

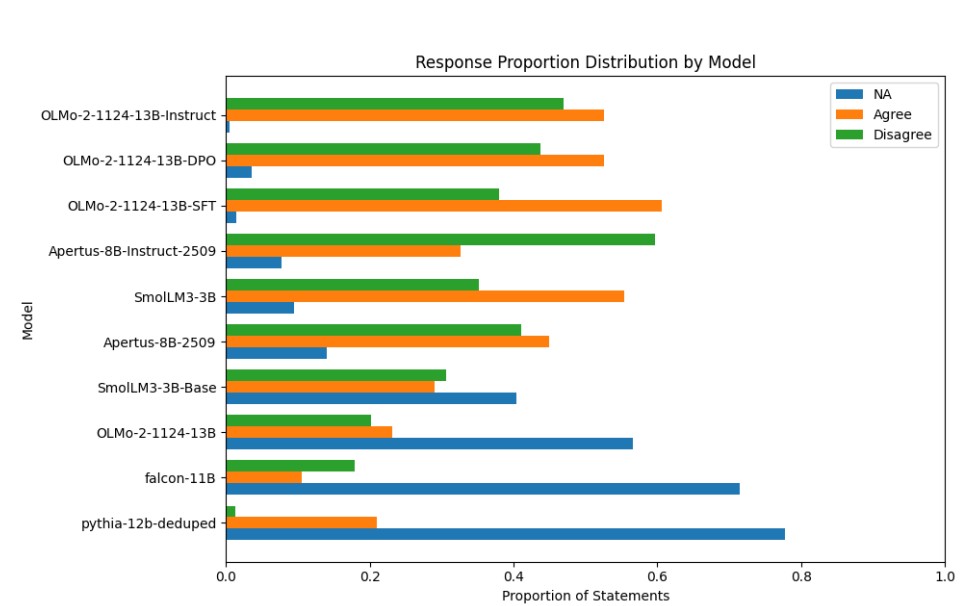

Figure 12: Distribution of answers across all models.

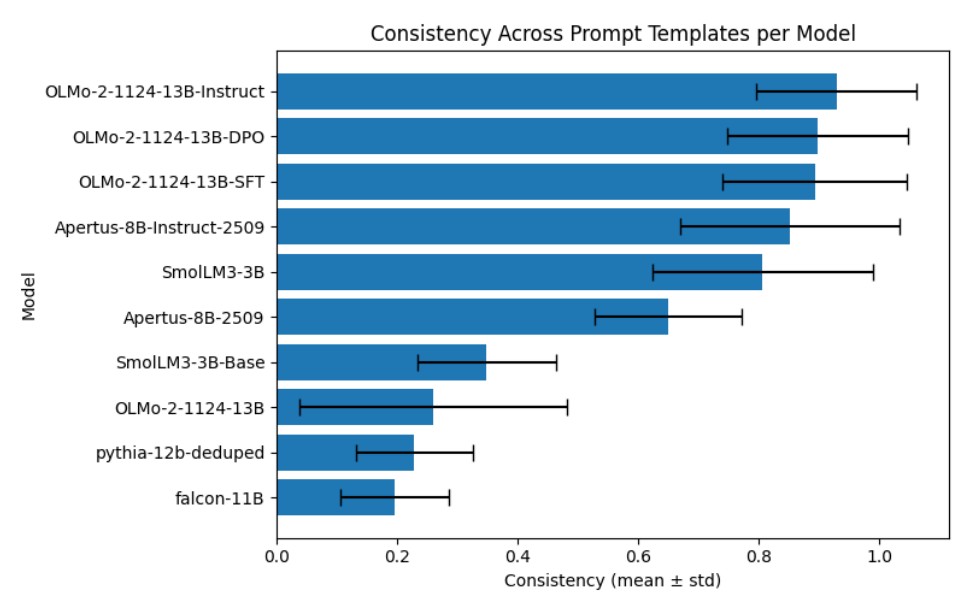

Figure 13: Consistency of answers (agree, disagree and NA) across templates.

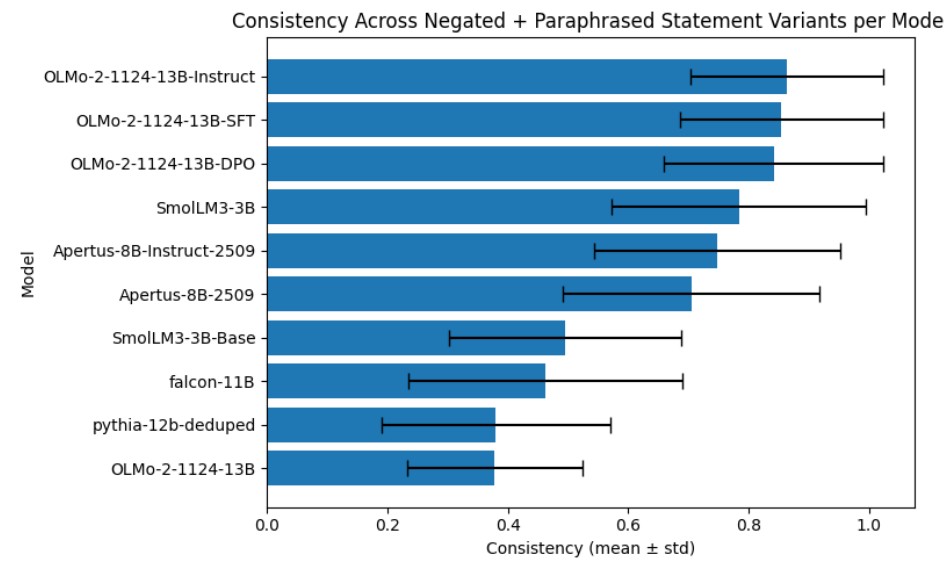

Figure 14: Consistency of answers (agree, disagree and NA) across paraphrases, negation and opposite versions of the statements.

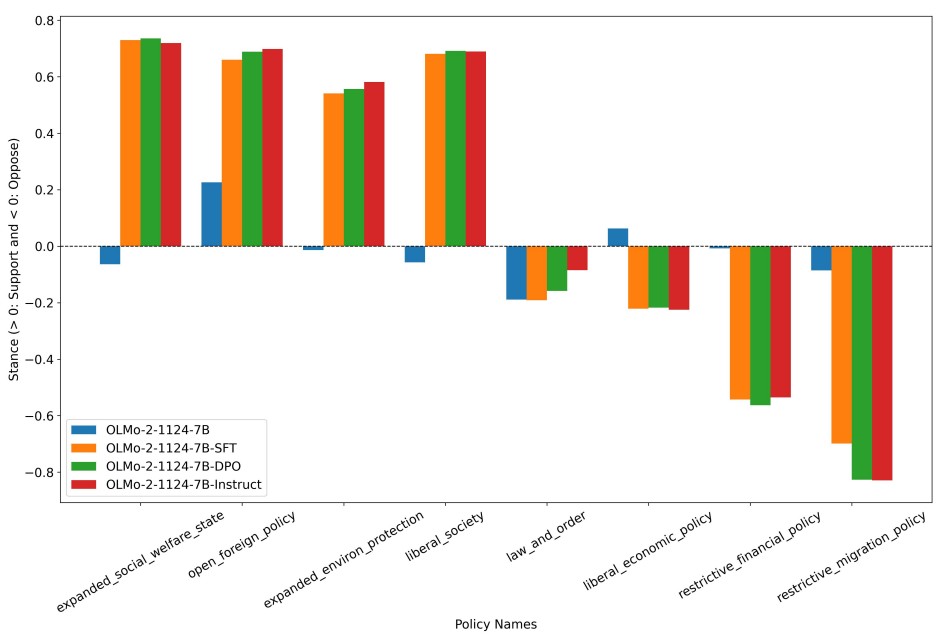

Figure 15: Political biases in the model OLMO2-7B.

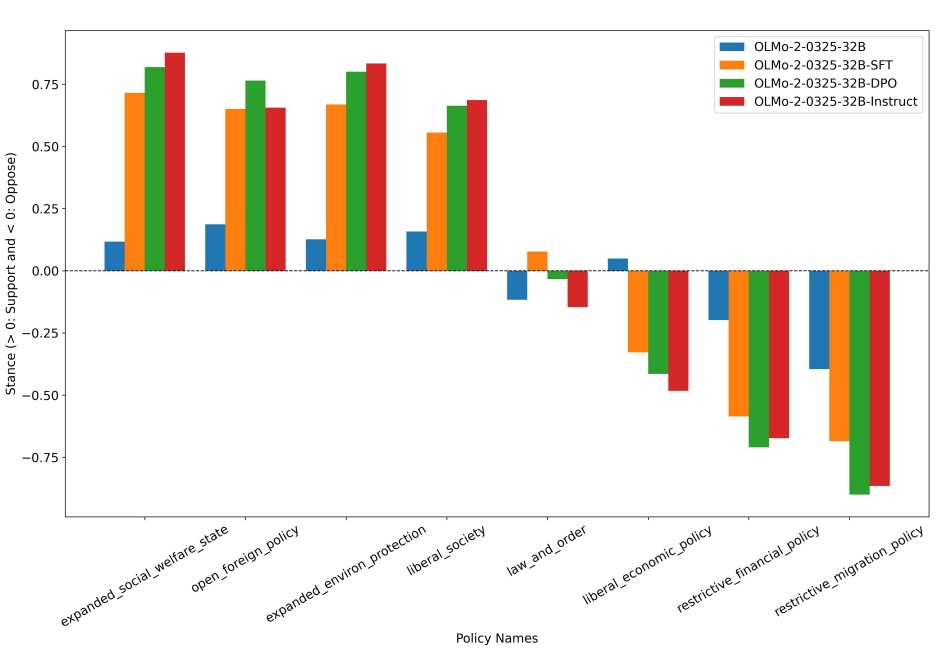

Figure 16: Political biases in the model OLMO2-32B.

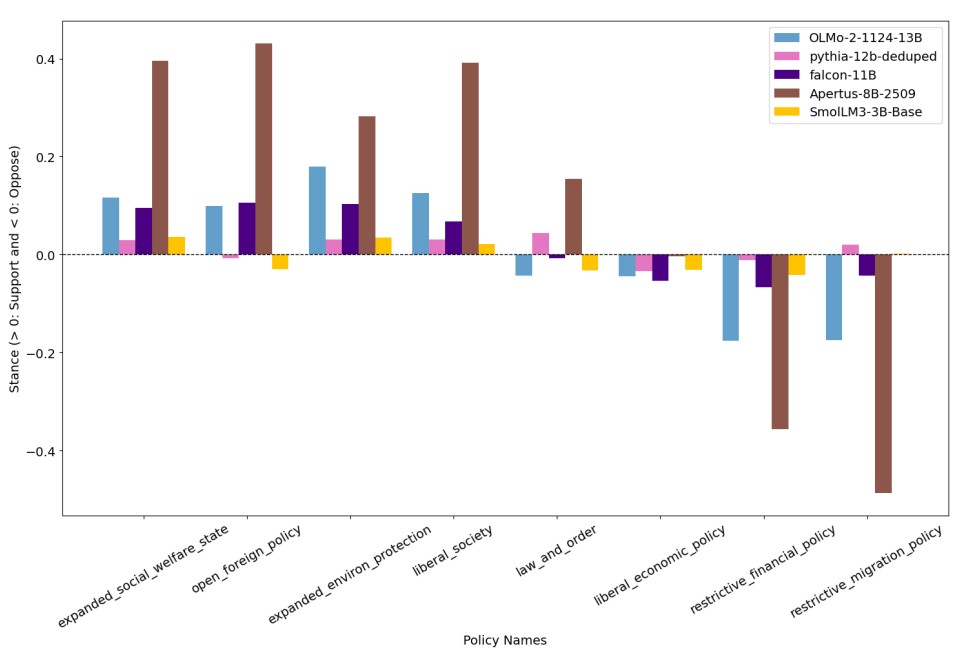

Figure 17: Stances of all evaluated pre-trained models.

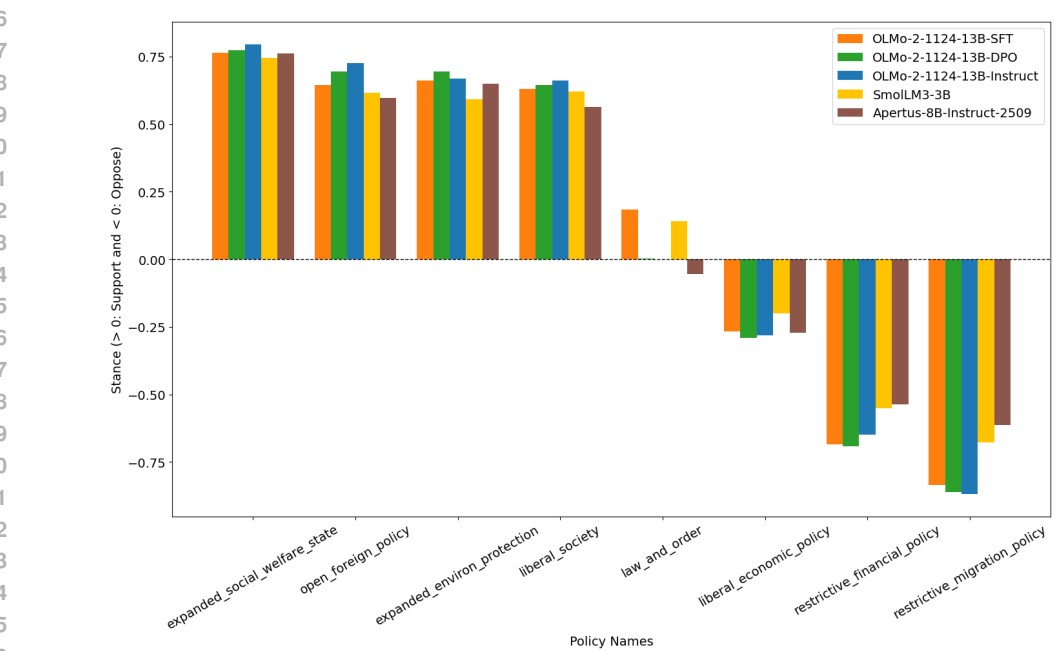

Figure 18: Stances of all evaluated post-trained models.

```
- (3) Restrictive Financial Policy
    Documents that support Restrictive Financial Policy predominantly:
{restrictive_financial_policy}
    Documents that oppose Restrictive Financial Policy predominantly:
{restrictive_financial_policy_oppose}
- (4) Law and Order
    Documents that support Law and Order predominantly:
{law_and_order}
    Documents that oppose Law and Order predominantly:
{law_and_order_oppose}
- (5) Restrictive Migration Policy
    Documents that support Restrictive Migration Policy predominantly:
{restrictive_migration_policy}
    Documents that oppose Restrictive Migration Policy predominantly:
{restrictive_migration_policy_oppose}
- (6) Expanded Environmental Protection
    Documents that support Expanded Environmental Protection predominantly:
{expanded_environ_protection}
    Documents that oppose Expanded Environmental Protection predominantly:
{expanded_environ_protection_oppose}
- (7) Expanded Social Welfare State
    Documents that support Expanded Social Welfare State predominantly:
{expanded_social_welfare_state}
    Documents that oppose Expanded Social Welfare State predominantly:
{expanded_social_welfare_state_oppose}
- (8) Liberal Society
    Documents that support Liberal Society predominantly:
{liberal_society}
    Documents that oppose Liberal Society predominantly:
{liberal_society_oppose}

# INSTRUCTIONS:
For each input text:
    1. Identify which of the above policy issues are discussed or implied by going through
    and evaluating the issues step by step.
    2. For each issue, classify the stance the text takes:
        - "neutral" if the issue is not mentioned, or it's mentioned, but the stance is
        ambiguous or not clearly expressed.
        - "support" if the text expresses approval, endorsement, or argument in favor of the
        issue.
        - "oppose" if the text expresses rejection, criticism, or argument against the issue.
    Only assign policy issues that are explicitly or strongly implied  the content.

# OUTPUT FORMAT:
{
```

Table 22: Portion of documents classified as *support* (sup), *oppose* (opp), and *neutral* (neu) by policy issue for each dataset.

| Dataset | Dolma | | | Dolmino | | | SFT-mix | | | DPO-mix | | |
|---|---|---|---|---|---|---|---|---|---|---|---|---|
| Stance | sup | opp | neu | sup | opp | neu | sup | opp | neu | sup | opp | neu |
| social welfare state | 0.32 | 0.04 | 0.63 | 0.39 | 0.06 | 0.55 | 0.37 | 0.01 | 0.62 | 0.43 | 0.02 | 0.55 |
| open foreign policy | 0.05 | 0.05 | 0.90 | 0.06 | 0.06 | 0.88 | 0.02 | 0.01 | 0.97 | 0.04 | 0.02 | 0.94 |
| environmental protection | 0.12 | 0.01 | 0.87 | 0.25 | 0.02 | 0.73 | 0.12 | 0.00 | 0.88 | 0.20 | 0.01 | 0.79 |
| liberal society | 0.44 | 0.06 | 0.50 | 0.48 | 0.06 | 0.47 | 0.62 | 0.01 | 0.37 | 0.52 | 0.03 | 0.45 |
| law and order | 0.11 | 0.16 | 0.73 | 0.07 | 0.15 | 0.78 | 0.03 | 0.04 | 0.93 | 0.06 | 0.07 | 0.88 |
| liberal economic policy | 0.05 | 0.13 | 0.82 | 0.07 | 0.18 | 0.75 | 0.03 | 0.03 | 0.94 | 0.06 | 0.06 | 0.88 |
| restrictive financial policy | 0.02 | 0.07 | 0.92 | 0.02 | 0.10 | 0.88 | 0.01 | 0.01 | 0.98 | 0.01 | 0.03 | 0.96 |
| restrictive migration policy | 0.03 | 0.07 | 0.90 | 0.02 | 0.08 | 0.90 | 0.01 | 0.06 | 0.93 | 0.02 | 0.06 | 0.93 |

Table 23: Results of the zero-shot classification per policy issue.

| category | majority macro-F1 | macro-F1 |
|---|---|---|
| restrictive-migration-policy | 0.31 | 0.83 |
| expanded-environ-protection | 0.31 | 0.78 |
| open-foreign-policy | 0.30 | 0.73 |
| expanded-social-welfare-state | 0.27 | 0.60 |
| restrictive-financial-policy | 0.28 | 0.60 |
| liberal-society | 0.27 | 0.58 |
| liberal-economic-policy | 0.28 | 0.57 |
| law-and-order | 0.30 | 0.53 |
| Average | 0.29 | 0.65 |

```
  "reasoning": "<your step-by-step reasoning about the classification>",
  "policy_stances": {
    "Open Foreign Policy": "neutral" | "support" | "oppose",
    "Liberal Economic Policy": "neutral" | "support" | "oppose",
    "Restrictive Financial Policy": "neutral" | "support" | "oppose",
    "Law and Order": "neutral" | "support" | "oppose",
    "Restrictive Migration Policy": "neutral" | "support" | "oppose",
    "Expanded Environmental Protection": "neutral" | "support" | "oppose",
    "Expanded Social Welfare State": "neutral" | "support" | "oppose",
    "Liberal Society": "neutral" | "support" | "oppose",
  }
}
Return your answers as a JSON object.

# ANSWER:
```

Results of the zero-shot classification and a majority baseline for comparison can be found in Table 23.

Table 22 presents the proportion of documents that have been classified as supporting, opposing or being neutral per policy issue. Documents can have multiple policy issue labels. We only classified the documents that had been classified as left and right. 17,434 documents in DOLMA; 13,911 in DOLMINO; 3,105 in SFT-MIX, and 6,712 in DPO-MIX.

Table **??** shows the Pearson correlation between the stances of the models and the stance found in the training documents when compared by policy issue.

## A.10 RESULTS FROM REFINEDWEB AND THE PILE

### A.10.1 SOURCE DOMAINS

The PILE does not have source domains available, therefore, our analysis are made for RefinedWeb. Figure shows the websites with the highest proportion of documents normalized by the count per

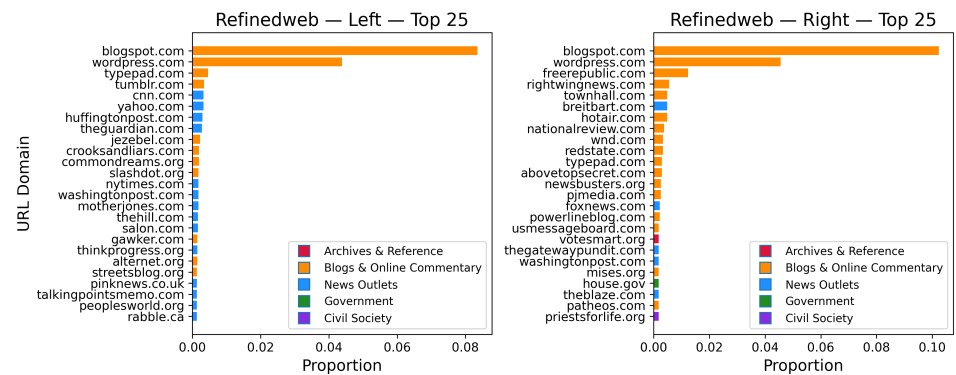

Figure 19: Top 30 source domains from RefinedWeb in comparison with Dolma and Dolmino.

Figure 20: Top 25 source domains from documents classified as left or right.

dataset sorted by the total count. Documents come from very similar sources. The Rank-Biased Overlap (RBO) between the datasets shows that DOLMA and REFINEDWEB have a very similar rank (RBO=0.765) while DOLMA and DOLMINO and DOLMINO and REFINEDWEB are moderately similar with RBO=0.522 and RBO=0.529 respectively.

Figure 20 shows the proportion of the top 25 domains found in the RefinedWeb documents. The tendency is still the same as in OLMO2. More news outlets are found in the left-leaning documents while more blogs and found in the right-leaning documents.

Figure 21: Most often topics in the pretraining datasets.

### A.10.2 TOPIC MODELING

Figure 21 shows the distribution of left and right documents for the 10 most largest clusters in the topic modeling. Confirming the findings in the source domains, results show that REFINEDWEB and THEPILE are most similar to Dolma in terms of frequent topics as well.

### A.11 LLM USAGE

We have used ChatGPT to aid in writing by rephrasing sentences we had already written. Moreover, we have utilized Grammarly to correct errors and provide suggestions for improving our writing.

