# OpenReview forum: "What Is The Political Content in LLMs' Pre- and Post-Training Data?"
_ICLR.cc/2026/Conference — ICLR 2026 Conference Withdrawn Submission_

### Official Review · Reviewer_ux9s · 2025-10-27

**Soundness:** 3
**Presentation:** 3
**Contribution:** 2
**Rating:** 4
**Confidence:** 4

**Summary:**

The paper investigates the political bias present in open-sourced LLM training datasets and correlates that to bias responses given by the OLMO2 model trained on these datasets. The paper first uses an LLM-based method to identify the political bias present in samples of the datasets. The paper then uses a combination of models and methods to analyze the differences in source domains and topics of the identified biased data. Finally, the paper prompts the OLMO2 model for its political viewpoints on topics and correlates these stances with the topical stances identified in the training datasets. The paper generally finds that the model inherits the political biases of the datasets that it is trained on.

**Strengths:**

The paper is attacking a significant problem, is clearly written, and has reasonably good validity for its testing procedures. As the paper points out, LLMs and LLM-driven technologies like AI Agents are increasingly being relied upon for truthful information, including in the political sphere. Furthermore, the paper presents a significant finding: even if the politically biased parts of the training data are in the minority overall, they appear to carry through in the generations produced by the LLM. The paper is also generally thorough in validating any claims it makes. For example, the paper tests its LLM-based classification of political bias on other benchmark bias datasets as well as the protocol established in Goet (2019). Another example comes in prompting the model for its political opinions on topics. The paper uses a number of different phrasings in the prompt and different proxies for the implicit political stance to really see if the political opinions of the model is stable. I also appreciate the thoroughness of the analysis of the differences between the sources of right-leaning and left-leaning documents in the datasets. Overall, the paper is generally thorough in its evaluation and has a significant result in finding a correlation between the biases present in the dataset and an LLM trained on that dataset.

**Weaknesses:**

The paper does lack some novelty and grounding in other works that have investigated political bias in LLMs as well as concepts like stance or bias labeling by LLM. For the latter, authors like Rozado in “The Political Preferences of LLMs” and Fujimoto and Takemoto “Revisiting the Political Biases of ChatGPT” used political preference tests as a way to measure the bias of models. For stance labeling by LLM, there are a number of techniques that have been developed for this particular task, including with testing on politically sensitive stance datasets. Some examples include  Zhang et al. “A Logically Consistent Chain-of-Thought Approach for Stance Detection”, Ziems et al. “Can Large Language Models Transform Computational Social Science?”, and Cruickshank and Ng “Prompting and Fine-Tuning Open-Sourced Large Language Models for Stance Classification,” to name a few. The work would have better methodological grounding and validity (and likely see better scores against the benchmark testing that they do) if they adopted some already established approaches. Finally, there are other works that have found similar results to the significant result that this paper did, albeit by different methods. Ng et al. In “Examining the Influence of Political Bias on Large Language Model Performance in Stance Classification” found a statistically significant difference in the performance of LLMs across various politically oriented stance classification tasks and that this difference primarily manifests at the dataset level. Overall, I think the work is lacking somewhat in novelty and could benefit from adopting established approaches for tasks like bias or stance labeling by LLM.

**Questions:**

- Do these results extend to other forms of bias? I believe other works have also indicated that the training data gives rise to gender bias or regional biases in LLMs.

- Where is the correlation coefficient coming from in the last paragraph of section 6? Is it just the correlation of the 8 topical stance valences between the model’s outputs and the accumulation of the stance labeling done on the datasets?

---

> ### Author Response · Authors · 2025-11-21
> **Response**
>
> > The paper does lack some novelty and grounding in other works that have investigated political bias in LLMs as well as concepts like stance or bias labeling by LLM.
>
> In terms of novelty, our paper is the first to systematically analyse the political content of pre- and post-training data. For that, we need to use some well-established methods for text analysis like classification, topic modeling, and prompting for bias evaluation. We are not sure what the reviewer means by "lack of novelty".
>
> > For the latter, authors like Rozado in “The Political Preferences of LLMs” and Fujimoto and Takemoto “Revisiting the Political Biases of ChatGPT” used political preference tests as a way to measure the bias of models.
>
> True, there are by now many political tests that have been used to evaluate political biases in LLMs. The one that we have chosen for our paper has already been validated and has been published in a renowned journal of the association of computational linguistics. Therefore, we believe that our method is also valid for evaluating biases in LLMs. Moreover, the method we use is able to evaluate biases at a fine-grained level of policy issues, allowing us to have a more nuanced analysis of the biases in models and in the stance of the training documents. Finally, the objective of our paper is not to offer a new framework or dataset for bias evaluation, but it is to investigate political content in LLM’s training data and how it correlates with model’s behavior.
>
> > For stance labeling by LLM, there are a number of techniques that have been developed for this particular task, including with testing on politically sensitive stance datasets.
> Some examples include Zhang et al. “A Logically Consistent Chain-of-Thought Approach for Stance Detection”, Ziems et al. “Can Large Language Models Transform Computational Social Science?”, and Cruickshank and Ng “Prompting and Fine-Tuning Open-Sourced Large Language Models for Stance Classification,” to name a few...
>
> Thanks for pointing out these papers. What you’re pointing to are different methods for classifying the stance of documents. We believe we’re following the standard practices for stance detection. That is, we have a target T, in this case, a policy issue such as restrictive migration and a text X. Our classifier predicts whether X opposes, supports or is neutral towards T. We evaluate our classifier towards human annotations as described in Section 6.2 and Appendix A.8. Finally, we use zero-shot classification and the latter paper you recommend also highlights that models do not get better when fine-tuned, confirming our approach is suitable for the task.
>
> > Finally, there are other works that have found similar results to the significant result that this paper did, albeit by different methods. Ng et al. In “Examining the Influence of Political Bias on Large Language Model Performance in Stance Classification” found a statistically significant difference in the performance of LLMs across various politically oriented stance classification tasks and that this difference primarily manifests at the dataset level.
>
> Thanks for pointing out this paper. The focus of the paper is quite different from ours though. They evaluate the performance of stance classifiers and how biased classifiers are while our paper investigates the political content in LLM’s training data and correlates it with the model’s behavior.

---

### Official Review · Reviewer_Gm5a · 2025-10-28

**Soundness:** 2
**Presentation:** 2
**Contribution:** 3
**Rating:** 4
**Confidence:** 3

**Summary:**

This paper investigates how political biases in large language models originate from their training data. Using OLMO2, the largest open-source LLM with full pre- and post-training datasets, the authors sample documents, classify them as left-, right-, or neutral-leaning using a validated classifier, and analyze content, sources, and framing. Results show that left-leaning documents vastly outnumber right-leaning ones across datasets, and pre-training data contain more politically engaged material than post-training data. Moreover, models’ policy stances correlate strongly (r = 0.90) with the political leanings in their training corpora, indicating that political bias largely emerges during pre-training.

**Strengths:**

1. First systematic data-level study connecting political content in both pre- and post-training corpora to model behavior.

2. Uses OLMO2’s transparent datasets and a rigorous sampling plus validation pipeline.

3. Presents interpretable topic and framing analysis revealing nuanced ideological contrasts.

4. Strong empirical evidence (r=0.90) linking data stance to model bias, with clear implications for fairness and transparency.

**Weaknesses:**

1. The political classification model relies on heuristic assumptions about “neutral” data, potentially oversimplifying the ideological spectrum.

2. Focus limited to one model (OLMO2), so generalization to other LLMs is unclear.

3. The causal direction—whether bias truly originates from pre-training or is reinforced later—is not conclusively disentangled.

4. Some sections (e.g., topic clustering) depend heavily on automated summarization, which might introduce interpretive bias.

**Questions:**

1. How robust is the left-right classifier across cultural or linguistic contexts beyond U.S.-centric framing?

2. Could the observed correlation (r=0.90) arise partly from model alignment procedures rather than purely pre-training data?

3. How do you ensure that automated topic summaries or GPT-generated cluster labels do not reinforce the same ideological bias you are measuring?

4. Have you considered quantifying how much each training stage amplifies or attenuates the bias magnitude?

---

> ### Author Response · Authors · 2025-11-21
> **Response and further experiments running**
>
> We thank you for your comments and questions.
>
> > The political classification model relies on heuristic assumptions about “neutral” data, potentially oversimplifying the ideological spectrum.
>
> We understand that left and right simplified the ideological spectrum, however, we would like to point out that political scientists themselves have relied and rely until nowadays on this type of “dimensionality reduction” to investigate parties and politicians in text or speech [1,2,3]. Moreover, we ran an analysis of the neutral documents to check whether they include political content for robustness check and our results in appendix A.4 DOCUMENTS CLASSIFIED AS NEUTRAL show that they indeed do not include political content.
>
> > Focus limited to one model (OLMO2), so generalization to other LLMs is unclear.
>
> Please, refer to the detailed answer we've given to second point of reviewer w7Lr. We're running extra experiments for the generalization of results on other datasets.
>
> > The causal direction—whether bias truly originates from pre-training or is reinforced later—is not conclusively disentangled.
>
> This is true, unfortunately we can’t disentangle it with our study. Our paper focuses on investigating the political content in LLM’s training data. Our results show that the highest concentration of politically loaded content is in the pretraining rather than in the post-training data. For us to be able to disentagle this question completely, we would have to retrain models from scratch with a different data distribution. As of now, we’re not able to do that because of the amount of computational resources needed for that. Training a small generative model wouldn’t be helpful to answer this question as we’ve seen in our experiments that small models (1B) aren’t consistent in generating their answers and generate a lot of non-sense/not-related content when prompted with specific questions.
>
> > Some sections (e.g., topic clustering) depend heavily on automated summarization, which might introduce interpretive bias.
>
> While it’s true that this section relies on automated summarization, it’s also true that this part of the experiments is not conclusive for our questions. Section 5.2 is a complementary analysis to understand the type of content present in the left and right classified documents overall. All the other experiments have been evaluated with human annotations and in domain data.
>
> Questions:
>
> > How robust is the left-right classifier across cultural or linguistic contexts beyond U.S.-centric framing?
>
> It’s a very good question to consider when we, in future work, evaluate non-English data as well. For now, most of the English data comes from countries which have a similar left-right political ideology and the largest part of this English data comes from the US (also proven by the domains where the documents mostly come from in Figure 1). When adding the analysis of other languages, we believe it’ll be necessary to include human annotations to evaluate the robustness of our classifier.
>
> > Could the observed correlation (r=0.90) arise partly from model alignment procedures rather than purely pre-training data?
>
> Table 21 in appendix shows a correlation of 0.88 between the BASE model and DOLMA and a correlation of 0.91 between OLMO2 and DOLMINO, showing that there is a high correlation also between the BASE model and the pretraining datasets (DOLMA and DOLMINO).
>
> > How do you ensure that automated topic summaries or GPT-generated cluster labels do not reinforce the same ideological bias you are measuring?
>
> We believe that if there is any type of bias, it was minimized by the experimental setup. We did not prompt the model with the political leaning of the documents, only with the documents themselves to make sure we don’t bias the summaries of the models.
>
> > Have you considered quantifying how much each training stage amplifies or attenuates the bias magnitude?
>
> Yes, we considered doing that during our experiments. However, we checked the code both from OLMO2 and from APERTUS and even though the checkpoints are available, the data order used during training is not traceable, therefore, we are not able to run such experiments.
>
> [1] Jolly, S., Bakker, R., Hooghe, L., Marks, G., Polk, J., Rovny, J., Steenbergen, M., and Vachudova, M. A. (2022). Chapel Hill expert survey trend file, 1999–2019. Electoral Studies, 75:102420.
>
> [2] Burst, T., Krause, W., Lehmann, P., Lewandowski, J., Matthieß, T., Merz, N., Regel, S., and Zehnter, L. 208 Bibliography (2021). Manifesto corpus. version: 2021.1. Berlin: WZB Berlin Social Science Center.
>
> [3] Benjamin E. Lauderdale and Alexander Herzog. 2016.Measuring Political Positions from Legislative Speech. Political Analysis, 24(3):374–394.

---

> > ### Author Response · Authors · 2025-12-02
> > **Results on Pythia and Falcon**
> >
> > The results of our further experiments that check for the generalization of our findings in other models/datasets can be found above in the last response to Reviewer w7Lr.

---

### Official Review · Reviewer_w7Lr · 2025-11-03

**Soundness:** 2
**Presentation:** 3
**Contribution:** 2
**Rating:** 2
**Confidence:** 4

**Summary:**

The paper conducts a comprehensive analysis of the political content of pre- and post-training data used in LLMs, focusing on the open-source OLMO2 model. They show that left-leaning documents predominate across datasets, with pre-training corpora containing significantly more politically engaged content than post-training data. Moreover, the author provides a dataset including training documents automatically annotated as left-leaning, right-leaning, or neutral.

**Strengths:**

1.Systematically analyzes the political content in the dataset used to train the OLMO2 model.
2. Provides a dataset containing both left- and right-leaning documents.
3. First work to focus on political bias in training data rather than model outputs.

**Weaknesses:**

1. The paper mainly provides a comprehensive analysis of the political issues in the training corpus but does not examine the influence of LLM, particularly regarding its mechanism, or propose any methods to address this issue.
2. All analysis is only based on the training corpus from the OLMO2 model, but the paper only analyzes data not including the model's response; it lacks a comprehensive cross-model or cross-corpus comparison, which may introduce bias.
3. No sufficient analysis of the classifier model and no human-in-the-loop evaluation for generated content

**Questions:**

1. Did the authors do any human evaluations to make sure that the political-leaning classifications in the corpus were correct? If not, how was the automatic classifier's reliability checked?
2. What do the authors think about ways to reduce bias during the data-curation stage?

**Details Of Ethics Concerns:**

The paper explicitly studies political bias (right- and left-leaning) in training data.

---

> ### Author Response · Authors · 2025-11-21
> **Response with more experiments running now to be reported later**
>
> We thank you for your comments.
>
> > The paper mainly provides a comprehensive analysis of the political issues in the training corpus...
>
> We would like to emphasize that the main objective paper is not to understand how the model would behave differently with a different training regime (e.g. change in data), but to investigate the political content contained in training data, which has never been studied before and which we believe must be the first step to understanding what can be changed or taken into consideration for building the training data of models. Other papers have investigated how post-training can steer models to different political opinions [1] or political positioning (left vs right) [2]. Finally, pre-training models from scratch with different sets of data distribution is not feasible because of the amount of computational resources needed.
>
> > All analysis is only based on the training corpus from the OLMO2 model,...
>
> It’s true that we have used only OLMO2 for our analysis and we understand that it is a limitation, it’s not true that we do not include the model’s responses as our entire Section 6 is exactly about that.
>
> Concerning the first point, we decided for OLMO2 because of two main reasons:
> (1) When we were running the experiments, it was the only fully open source model (weights, code and data) with training datasets and model checkpoints available for all the training stages (base, SFT, DPO, and Instruct), allowing us to have a more thorough analysis of the data in correlation with the models.
> (2) The training datasets from OLMO2 overlap largely with the training data from other open source models because they come from CommonCrawl. During this week, we have created the table below which documents the datasets that used to train different open source models.
>
> The table below shows the overlap in the training datasets.
>
> | Model | OLMO2 | Marin-8B | SmolLM | Apertus | Falcon2 | Pythia |
> |:------|:------|:----------|:--------|:---------|:---------|:--------|
> | Dolma | ✅ | ✅ | ❌ | ❌ | ❌ | ❌ |
> | Dolmino | ✅ | ✅ | ✅ | ❌ | ❌ | ❌ |
> | Overlap DCLM | ✅ | ✅ | ✅ | ✅ | ❌ | ❌ |
> | Overlab FINEWEB | ✅ | ❌ | ✅ | ✅ | ❌ | ❌ |
> | Pile | ❌ | ❌ | ❌ | ❌ | ❌ | ✅ |
> | C4 | ❌ | ❌ | ❌ | ❌ | ❌ | ❌ |
> | CommonCrawl | ❌ | ❌ | ❌ | ❌ | ❌ | ❌ |
> | Wikipedia | ✅ | ✅ | ✅ | ❌ | ❌ | ❌ |
> | ArXiv | ✅ | ✅ | ✅ | ❌ | ✅ | ❌ |
> | Books | ✅ | ❌ | ❌ | ❌ | ✅ | ❌ |
> | peS2o | ✅ | ✅ | ✅ | ❌ | ❌ | ❌ |
> | FLAN | ✅ | ✅ | ✅ | ❌ | ❌ | ❌ |
> | Nemotron-CC | ❌ | ✅ | ❌ | ❌ | ❌ | ❌ |
> | Proof-Pile-2 | ❌ | ✅ | ❌ | ❌ | ❌ | ❌ |
> | SmolLM corpus | ❌ | ❌ | ✅ | ❌ | ❌ | ❌ |
> | RefinedWeb | ❌ | ❌ | ❌ | ❌ | ✅ | ❌ |
> | Reddit | ❌ | ❌ | ❌ | ❌ | ✅ | ❌ |
> | HackerNews | ❌ | ❌ | ❌ | ❌ | ✅ | ❌ |
>
> We don’t report on the datasets related to Math, Reasoning, and Code. In the table, we observe that (OLMO2 and SmolLM) and (OLMO2 and Marin-8B)  overlap in 7 datasets, and (OLMO2 and Apertus) overlap in 2 datasets. However, note that they are very large datasets, and therefore, the overlap might be equivalent as well. And finally, Falcon only overlaps with ArXiv and Books while Pythia is the only model that doesn’t overlap with any of the datasets from OLMO2. After this thorough analysis, we decided to run further experiments with the datasets RefinedWeb and Pile from the models that are most different from OLMO2’s training data, Falcon and Pythia. We will post the results here as soon as possible.
>
> > No sufficient analysis of the classifier model and no human-in-the-loop evaluation for generated content.
>
> We would like to point to the sections where we explain the human evaluation of our classifiers. Appendix A.2 describes details of the manually datasets used in our evaluation and appendix A.3 shows the results of the evaluation of our left/right/neutral classifier. In total, we validate this classifier on three different manually annotated datasets (NEWSLEAN, POLSTANCE and US CONGRESS).
>
> The stance classifier has been evaluated on the StancePol dataset (described in Appendix A.2.1). A manually annotated dataset with documents extracted directly from Dolma for the annotations, and therefore, for in domain evaluation.
>
> Answers to questions:
>
> 1) We responded to this question above. Moreover, appendix A.2 discusses how similar the NEWSLEAN corpus is to the pretraining data (i.e. many domains from the annotated dataset with which we evaluate our classifier overlap with the domains found in the training data), showing the robustness of our human evaluation.
>
> 2) We discussed this question in the discussion section 7, page 9.
>
> [1] Stammbach, D., Widmer, P., Cho, E., Gu̇lçehre, Ç., & Ash, E. (2024, November). Aligning large language models with diverse political viewpoints. EMNLP 2024.
>
> [2] Weeber, F., Ceron, T., & Padó, S. (2025). Do Political Opinions Transfer Between Western Languages? An Analysis of Unaligned and Aligned Multilingual LLMs. arXiv:2508.05553.

---

> > ### Author Response · Authors · 2025-12-02
> > **Results on Pythia and Falcon**
> >
> > To understand whether our results in OLMO2 generalize across other training corpora and models, we ran our analysis on The Pile and RefinedWeb, the datasets used to train Pythia-12B and Falcon-11B respectively. As explained above, these are the only models trained with web-scraped datasets that do not overlap OLMO2's. We've uploaded a new version of the paper with the results.
> >
> > In a summary:
> > - RefinedWeb has a source-domain distribution closely resembling that of Dolma, reflected in a high rank-biased overlap (RBO = 0.765), and a moderate overlap with Dolmino (RBO = 0.529). Results in Figure 19 and 20 in Appendix 10. The Pile unfortunately doesn't have information about the source domains to allow for this analysis.
> > - The top ranked topic clusters are very similar across models. Results in Figure 21.
> > - The political biases found in Pythia (base) and Falcon (base) are very similar to OLMO-base (cf. Figure 3a) except for Restrictive Migration Policy where Pythia shows a slight tendency in the opposite direction.
> > - The political stances observed in the pre-training corpora Dolma and Dolmino and The Pile and RefinedWeb are highly correlated. Results in Figure 3b. Below is the correlation between their stances:
> >
> > | Dataset    | Dataset1 | Pearson r | p-value |
> > | ---------- | -------- | --------- | ------- |
> > | RefinedWeb | Dolma    | 0.996     | 0.000   |
> > | RefinedWeb | Dolmino  | 0.979     | 0.000   |
> > | Pile       | Dolma    | 0.998     | 0.000   |
> > | Pile       | Dolmino  | 0.984     | 0.000   |
> > | Dolma      | Dolmino  | 0.991     | 0.000   |
> >
> > These results show that pre-training corpora coming from different datasets have very similar properties in terms of political content even though they have been subjected to different curation and filtering strategies. Finally, results also show that our interpretation of results generalize across corpora and models.

---

### Official Review · Reviewer_SMjd · 2025-11-11

**Soundness:** 3
**Presentation:** 3
**Contribution:** 3
**Rating:** 4
**Confidence:** 3

**Summary:**

This paper presents a thorough analysis of political content in pre- and post-training corpora of OLMO2. The authors find that left-leaning documents are more popular in all the analyzed datasets, and the predominant stance in the training data is also highly correlated with the models' political biases when evaluated on policy issues.

**Strengths:**

1. A thorough analysis of political content in LLMs' training datasets
2. The findings reveal how political content in the training data is potentially be converted into models' political bias.

**Weaknesses:**

1. One of the key weaknesses of this paper is the lack of direct implications for model training. Political content might also be connected to the quality of the training data. For example, the New York Times data is left-leaning but has also been recognized as a unique dataset with very high quality. The trade-off between data quality and political content is underexplored in the current draft.

2. While I really like the analysis of models' political biases in different training stages, it would be better if the author could conduct some experiments to further strengthen the causal relationships here. For example, try to change the data distribution of the DPO data and fine-tune the model in different settings. This will allow you to make stronger claims about the connection between data and model biases.

**Questions:**

N/A

---

> ### Author Response · Authors · 2025-11-21
> **Response**
>
> We thank you for your comments.
> > One of the key weaknesses of this paper is the lack of direct implications for model training. Political content might also be connected to the quality of the training data. For example, the New York Times data is left-leaning but has also been recognized as a unique dataset with very high quality. The trade-off between data quality and political content is underexplored in the current draft.
>
> Given that we are classifying the left and right documents in our dataset, we assume that only the documents that are more politically loaded are classified as left and right. Therefore, this would not influence the total quality of the training data. Moreover, we checked the number of times nytimes.com appears in the left/right documents vs neutral documents. We find that only 17,7% and 16,1% of the documents coming from the nytimes have been labelled as left/right while in Dolmino and Dolma respectively, showing that the proportion of politically loaded documents is very low.
>
> > While I really like the analysis of models' political biases in different training stages, it would be better if the author could conduct some experiments to further strengthen the causal relationships here. For example, try to change the data distribution of the DPO data and fine-tune the model in different settings. This will allow you to make stronger claims about the connection between data and model biases.
>
> We would like to emphasize that the main objective of this paper is not to understand how models behave differently with a different training regime (e.g. change in data), but to investigate the political content contained in training data, which has never been studied before. We believe that this is the first step to understanding, for example, what training data to include or not for example. Our results help inform what can be changed in the training data regime in order to obtain a different type of bias or towards building more impartial models.
> Moreover, other papers have already investigated how post-training steers models to different political opinions [1] or political positioning (left vs right) [2]. In terms of pretraining, unfortunately pretraining models from scratch with a different data distribution is not feasible due to the computational resources needed for that.
>
> [1] Stammbach, D., Widmer, P., Cho, E., Gu̇lçehre, Ç., & Ash, E. (2024, November). Aligning large language models with diverse political viewpoints. In Proceedings of the 2024 Conference on Empirical Methods in Natural Language Processing (pp. 7257-7267).
>
> [2] Weeber, F., Ceron, T., & Padó, S. (2025). Do Political Opinions Transfer Between Western Languages? An Analysis of Unaligned and Aligned Multilingual LLMs. arXiv preprint arXiv:2508.05553.

---

### Note · Authors · 2025-12-03

**Comment:**

We are grateful for the reviewers’ time. We have decided to withdraw our submission as the OpenReview leakage issue prevented us from having a continued discussion during the rebuttal period.

**Withdrawal Confirmation:**

I have read and agree with the venue's withdrawal policy on behalf of myself and my co-authors.